# Illuminating the function of the orphan transporter, SLC22A10, in humans and other primates

Sook Wah Yee [1], Luis Ferrández-Peral [2], Pol Alentorn-Moron [2],
Claudia Fontsere [2,3], Merve Ceylan [4], Megan L. Koleske[1], Niklas Handin[4],
Virginia M. Artegoitia[5], Giovanni Lara [1], Huan-Chieh Chien[1], Xujia Zhou[1],
Jacques Dainat [6], Arthur Zalevsky [1], Andrej Sali [1,7,8], Colin M. Brand[9,10],
Finn D. Wolfreys[11], Jia Yang [1], Jason E. Gestwicki [7,12], John A. Capra [1,9,10,13],
Per Artursson [4,14], John W. Newman [5,15], Tomàs Marquès-Bonet [2,16,17,18] &
Kathleen M. Giacomini [1,13] ✉

SLC22A10 is an orphan transporter with unknown substrates and function. The goal of this study is to elucidate its substrate specificity and functional characteristics. In contrast to orthologs from great apes, human SLC22A10, tagged with green fluorescent protein, is not expressed on the plasma membrane. Cells expressing great ape SLC22A10 orthologs exhibit significant accumulation of estradiol-17β-glucuronide, unlike those expressing human SLC22A10. Sequence alignments reveal a proline at position 220 in humans, which is a leucine in great apes. Replacing proline with leucine in SLC22A10-P220L restores plasma membrane localization and uptake function. Neanderthal and Denisovan genomes show proline at position 220, akin to modern humans, indicating functional loss during hominin evolution. Human SLC22A10 is a unitary pseudogene due to a fixed missense mutation, P220, while in great apes, its orthologs transport sex steroid conjugates. Characterizing SLC22A10 across species sheds light on its biological role, influencing organism development and steroid homeostasis.

About 30% of the members of the large Solute Carrier (SLC) Superfamily in the human genome have no known substrate[1], representing a major gap in understanding human biology. Deorphaning is the process of determining the function of a protein that has not yet been characterized. For deorphaning proteins in the SLC superfamily, which includes multi-membrane spanning transporters, phylogenetic analysis represents the first step for identifying the substrates of an orphan transporter. Other methods include metabolomic methods in cells or in knockout mice[2–4]. For example, the substrate of mouse SLC16A6, a transporter in the MonoCarboxylate Transporter Family, MCT7, was discovered through the analysis of amino acid transport in cell lines that overexpressed MCT7[5].

There has been increasing interest in deorphaning solute carrier transporters due to the significant potential role of SLCs in human physiology[1,6]. While some orphan genes in the SLC superfamily encode proteins which have evolved other functions and do not participate in transmembrane solute flux, it is more probable that these multi-membrane spanning proteins primarily serve as transporters. Therefore, efforts to deorphan SLC family members should include attempts to identify their endogenous substrates, ligands of the transporter that are translocated across biological membranes. In certain cases, the roles of transporters have been discovered, yet the substrate of the transporter remains unknown. This creates gaps in our mechanistic understanding of the transporter's function in those processes. For

example, Ren et al. used untargeted metabolomics and found elevated levels of lipid diacylglycerol and altered fatty acid metabolites in liver and plasma samples of Mct6 knockout mice[7]. This finding supports a role of SLC16A5 in lipid and amino acid homeostasis, but does not reveal its substrates and as such, the mechanism remains poorly understood. Similarly, the function of the orphan transporter SLC38A10 (SNAT10) was assessed by studying mice lacking the *Slc38a10* gene (Slc38a10-deficient mice). The findings indicated that *Slc38a10*-deficient mice exhibited reduced body weight and lower plasma levels of threonine and histidine. However, no study has specifically investigated whether these amino acids serve as substrates for SLC38A10[8]; therefore a gap in understanding the mechanism by which the transporter affects body weight remains. Information regarding recently deorphaned transporters is presented in a recent review[9].

Orphan transporters can be found in over 20 families in the SLC superfamily[1]. In the Solute Carrier 22 family A (SLC22A), there are 23 members that transport organic ions including 6 that are orphans[6]. Largely representing plasma membrane transporters, members of the SLC22A family are clustered together based on their charge specificity for organic cations (OCTs), organic anions (OATs), and organic zwitterion/cations (OCTNs). Solute carrier 22 family member 10 (SLC22A10) and its direct species orthologs are orphan transporters whose substrates and transport mechanisms are yet to be characterized. In humans, SLC22A10 has been given a protein name of OAT5. Based on Northern blotting[10] and RNA seq studies (https://www.proteinatlas.org/ENSG00000184999-SLC22A10/tissue)[11], human SLC22A10 is expressed specifically in the liver.

Orthologs of human SLC22A10 are present in many primates including great apes (https://useast.ensembl.org/Homo_sapiens/Gene/Compara_Ortholog?db=core;g = ENSG00000184999;r = 11:63268022-63311783). Intrigued by this observation, our study aims to identify the substrates of SLC22A10 and the transport mechanism by expressing primate orthologs of SLC22A10 in cell lines and performing analytical procedures including cellular uptake studies, metabolomic analyses and proteomic assays. We attempt to identify crucial amino acids that contribute to the differences in function between direct species orthologs in humans and great apes. Kinetic parameters and transport mechanisms of various predicted isoforms of SLC22A10 are determined, along with their ability to accumulate different endogenous ligands. Proteomic studies in cell lines recombinantly expressing human and chimpanzee SLC22A10 are conducted. Our study shows that human SLC22A10 was inactivated by a single missense mutation and is a unitary pseudogene. The ORF-disrupting mutation in SLC22A10, which led to Pro220, is not observed in great apes and primates. This particular amino acid is crucial for protein abundance and expression on the plasma membrane. Our work provides a roadmap for how orthologous genes, along with sequence comparison, and proteomic and transporter assays, can be used to deorphan the function of solute carrier proteins. These discoveries have significant evolutionary implications.

## Results

### Human SLC22A10 showed no expression on the plasma membrane and no transporter activity of prototypical anionic substrates of SLC22 family members

Human SLC22A10 is in a cluster that includes known organic anion transporters: SLC22A24 is closest, followed by SLC22A9, SLC22A11 and SLC22A12 (Fig. 1A). Phylogenetic analyses reveals that its closest homolog is SLC22A24. The substrates of SLC22A24 are steroid conjugates, bile acids and dicarboxylic acids, which our laboratory has successfully deorphaned[3]. Overexpression of human SLC22A10 tagged with GFP in the N-terminal resulted in no detection of a GFP-tagged protein on the plasma membrane (Fig. 1B). Furthermore, no uptake of prototypical organic anions was observed in cells expressing SLC22A10 whereas significant uptake was observed in cells expressing

known SLC22 organic anion transporters including SLC22A6, SLC22A8 and SLC22A24 (Fig. 1C).

### The long isoform of chimpanzee and gorilla SLC22A10 was expressed on the plasma membrane whereas the short isoform was not

The organic anion transporters depicted in Fig. 1A consist of 536 to 563 amino acid proteins. Predictions from reliable sources such as Uniprot, Ensembl, and NCBI Nucleotide databases confirmed that the human SLC22A10 gene produces a 541 amino acid isoform. Conversely, according to reports from Ensembl and UniProt, the orthologs of SLC22A10 found in great apes are predicted to have two isoforms: a short isoform comprising 540 amino acids and a longer isoform containing 552 amino acids. Because there was no detectable expression of the human SLC22A10 on the plasma membrane of cells recombinantly expressing the transporter (Fig. 1B), we inquired whether the direct species orthologs in great apes exhibited a similar lack of plasma membrane expression when expressed recombinantly in cells. In fact, we observed that the long isoforms (552 amino acids) of both chimpanzee and gorilla SLC22A10 were detected on the plasma membrane (Fig. 2A). The shorter isoforms (540 amino acids) of chimpanzee, bonobo and gorilla SLC22A10 showed a similar lack of plasma membrane localization as the human ortholog, which consists of 541 amino acids (Figs. 1A and 2A).

### Chimpanzee and gorilla SLC22A10 expressing the long isoform transport estradiol glucuronide but not other anions that are canonical substrates of members in the SLC22A family

Because the long forms of the great ape SLC22A10 showed a plasma membrane localization, we attempted to identify substrates of SLC22A10 using isotopic uptake assays in cells recombinantly expressing the long isoforms of the great ape transporters. Typical anions that are canonical substrates of members in the SLC22A family were screened for accumulation in human, chimpanzee, bonobo and gorilla expressing the long as well as the short isoforms. Significant accumulation of [³H]-estradiol-17β-glucuronide and [³H]-androstane-diol-3α-glucuronide were observed in cells expressing chimpanzee and gorilla SLC22A10 encoding the long but not the short isoforms (Fig. 2B, Supplementary Fig. 1A). No significant uptake was detected for other anions that are canonical substrates of members of the SLC22A family, such as estrone sulfate, taurocholic acid, cGMP, uric acid and succinic acid (Supplementary Fig. 1A–F). However, there was a small but significant uptake of [³H]-methotrexate in HEK293 cells expressing chimpanzee and gorilla SLC22A10 long isoforms (Supplementary Fig. 1G).

### The SLC22A10 protein in humans consists of 541 amino acids, resulting from a single nucleotide insertion that causes a frameshift in the last exon

The genetic mechanism that led to the formation of the 541 amino acid SLC22A10 protein in humans was investigated. Sequence alignments of the last exon (exon 10) of the SLC22A10 gene was compared between humans and great apes and revealed an insertion of one nucleotide leading to the expression of different isoforms in each species (see Fig. 2C). In particular, humans exhibit an A nucleotide insertion at the first base pair of exon 10, which is highly prevalent with an allele frequency of 98% in all populations in gnomAD[12] (11-63311006-GA-G (GRCh38)/11-63078478-GA-G (GRCh37)). The frequency ranges from 95% in the African population to over 99% in other ethnic groups. In contrast, the adenosine insertion has a 2.5% allele frequency in chimpanzees and is not present in other great apes. The adenosine insertion in the human SLC22A10 gene causes a frameshift and results in a 541 amino acid protein, instead of the predicted 552 amino acids in the SLC22A10 gene of great apes and in the humans who do not harbor the adenosine insertion (Fig. 2C). We utilized our previously generated

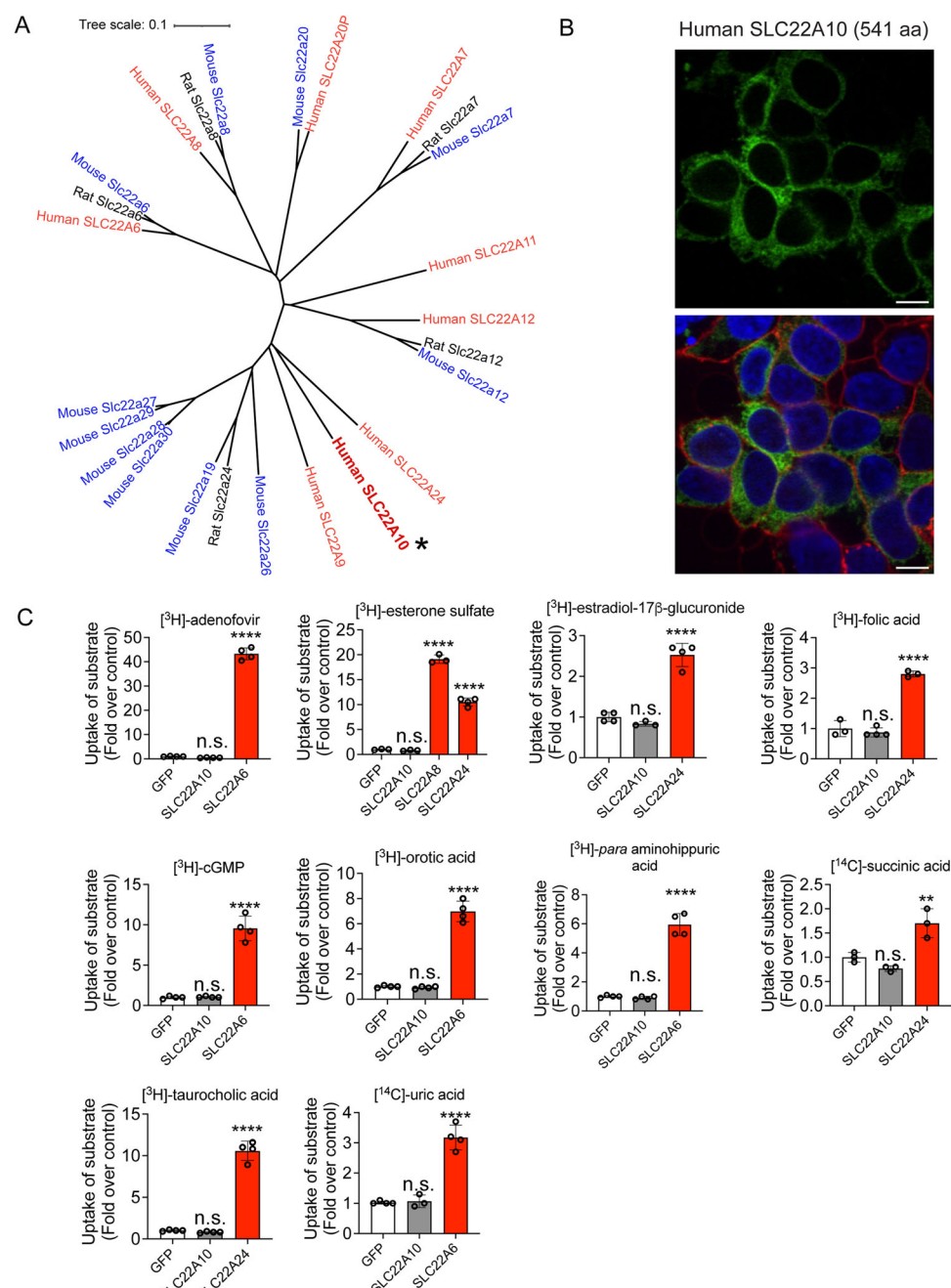

**Fig. 1 | Analysis of the phylogenetic tree, plasma membrane expression of SLC22A10, and uptake of organic anion substrates of the human SLC22 family.** **A** Multiple sequence alignments were performed with reference amino acid sequences for each anion transporter from humans and rodents, using the Clustal Omega Multiple Sequence Alignment program (https://www.ebi.ac.uk/Tools/msa/clustalo/). The dendrogram was generated from the output of the Clustal Omega alignment. Refer to the Source Data file to access the amino acid sequences for each transporter in this tree. * Human SLC22A10. **B** Localization of human SLC22A10 conjugated to green fluorescent protein (GFP) was examined in HEK293 cells using high-content imaging and cellular staining with the plasma membrane marker wheat germ agglutinin (WGA). Blue: DNA stain Hoechst marks the cell nucleus; Red: Plasma membrane marker WGA; Green: SLC22A10. Yellow: Merge. The results showed no colocalization of GFP-tagged SLC22A10 with WGA. The figure shows a representative image from two technical replicates. Scale bar: 10 μM. **C** Uptake of

various radiolabeled organic anions, which are typical substrates of organic anion transporters in the SLC22A family, was assessed. Uptake was performed 48 hours after transient transfection of plasmids encoding human SLC22A10, GFP expression vector, and one other member in the SLC22A family as a positive control. Accumulation of substrates inside cells was determined after 15 minutes. The scatter plot with bars shows the fold uptake of the substrate relative to the negative control. HEK293 Flp-In cells transiently transfected with the GFP vector served as the negative control. The plot displays the mean +/− standard deviation of three or four technical replicates ($n = 1$ shown as a representative experiment). Source data are provided as a Source Data file. Statistical significance was determined using a one-way analysis of variance (ANOVA) with Dunnett's multiple comparison test. n.s.: non-significant, **$p$ value < 0.01, ****$p$ value < 0.001. Similar results were obtained in two independent experiments.

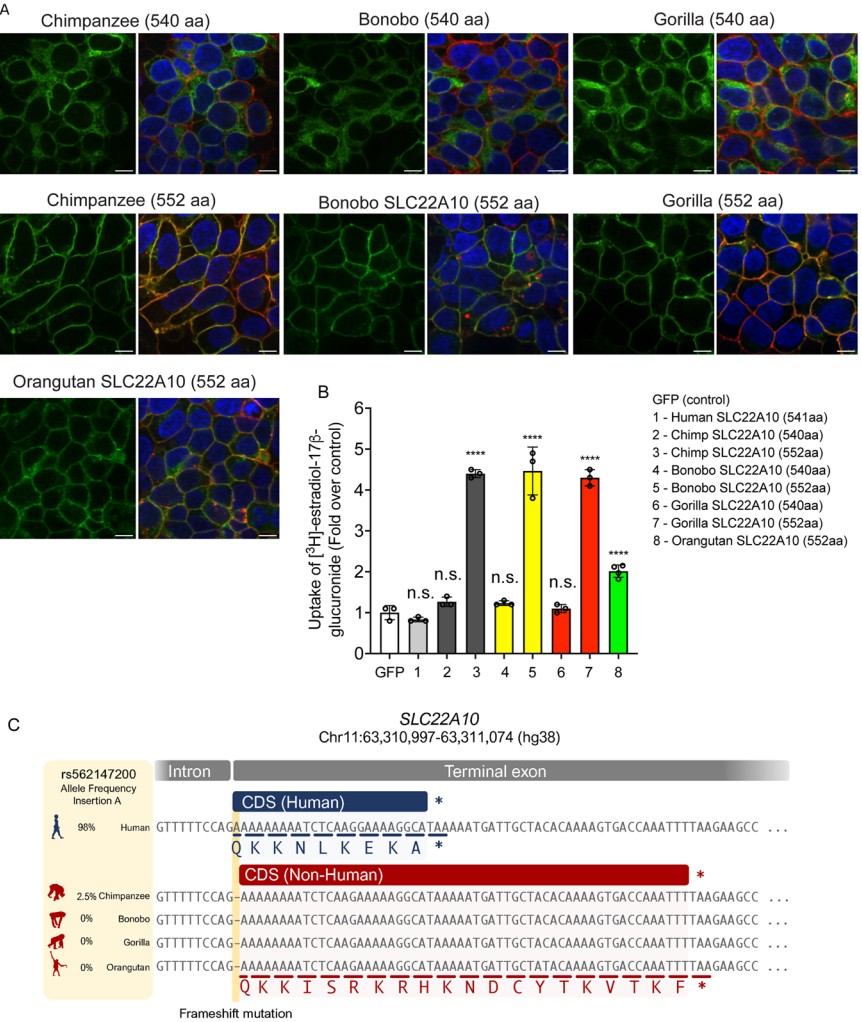

**Fig. 2 | Localization to the plasma membrane, uptake, and sequence comparison of human SLC22A10 were examined in comparison with SLC22A10 from great apes (chimpanzee, bonobo, gorilla and orangutan). A** The plasma membrane localization of SLC22A10 orthologs from great apes, which were conjugated to green fluorescent protein (GFP) in HEK293 Flp-In cells. The GFP tag is located at the N-terminus of SLC22A10. Confocal imaging revealed that the 552 amino acid isoforms of SLC22A10 from chimpanzee, bonobo, gorilla, and orangutan primarily colocalized with wheat germ agglutinin (WGA) on the plasma membrane of the cell. In contrast, the 540 amino acid isoform of SLC22A10 from bonobo, chimpanzee, and gorilla showed no colocalization of GFP-tagged SLC22A10 with WGA on the plasma membrane, suggesting intracellular localization in the cytoplasm. Blue: DNA stain Hoechst marks the cell nucleas; Red: Plasma membrane marker WGA; Green: SLC22A10. Yellow: Merge. The image shows a representative image from two technical replicates. Scale bar: 10 μM. **B** The uptake of [³H]-estradiol-17β-glucuronide was determined in HEK293 Flp-In cells overexpressing either a GFP expression vector or SLC22A10 expression vectors containing sequences from various primates including human, chimpanzee, bonobo, gorilla, and orangutan. SLC22A10 orthologs from chimpanzee, bonobo, gorilla, and orangutan expressing the longer isoform (552 amino acids) significantly accumulated [³H]-estradiol-17β-glucuronide. The scatter plot with bars shows the fold uptake of [³H]-estradiol-17β-glucuronide relative to the negative control. HEK293 Flp-In cells transiently transfected with the GFP vector served as the negative control. The plot displays the mean +/- standard deviation of three or four technical replicates (n = 1 shown as a representative experiment). Source data are provided as a Source Data file. Statistical significance was determined using a one-way analysis of variance (ANOVA) with Dunnett's multiple comparison test. n.s.: non-significant, ****p value < 0.001. Similar results were obtained in three independent experiments. **C** Sequence alignments of the last exon of SLC22A10 in human, chimpanzee, bonobo, gorilla, and orangutan are shown. In humans, the frequency of the A-allele insertion is significantly greater (98%) than in chimpanzees (2.5%) and is not present in available sequences from bonobos, gorillas, or orangutans. The A-allele insertion results in the expression of human SLC22A10 with 541 amino acids, while bonobo, gorilla, orangutan and the majority of chimpanzees are predicted to express isoforms of SLC22A10 with 552 amino acids. Chimpanzee image [https://www.phylopic.org/images/2f7da8c8-897a-445e-b003-b3955ad08850/pan-troglodytes], under license [https://creativecommons.org/licenses/by/3.0/], credit to T. Michael Keesey (vectorization) and Tony Hisgett (photography). Orangutan image [https://www.phylopic.org/images/67144c22-93c2-4dc0-ba13-9f9dd2d223b9/pongo-abelii], under license [https://creativecommons.org/licenses/by/3.0/], credit to Gareth Monger. All other silhouette images come from www.phylopic.org and are public domain images. All images were modified to different colors.

human and liver RNAseq data[13] alongside a large human RNA-seq databases (recount3[14]) to validate the splicing event in human and chimpanzee SLC22A10. Our analysis confirmed that splicing mostly occurs at the exact orthologous genomic region in both species, utilizing the canonical splice sites in their corresponding genomes. Thus, we found no coordinated splicing alterations that compensate for the A nucleotide insertion. Consequently, the additional A nucleotide

remains present in the final transcribed transcript in humans. This additional nucleotide provides evidence that human SLC22A10 protein contains 541 amino acids, while the chimpanzee SLC22A10 protein comprises 552 amino acids.

Additionally, SLC22A10 harbors a prevalent nonsense variant, p.Trp96Ter (rs1790218), which is frequently observed at high allele frequencies in human populations ranging from ~20% in African to 50%

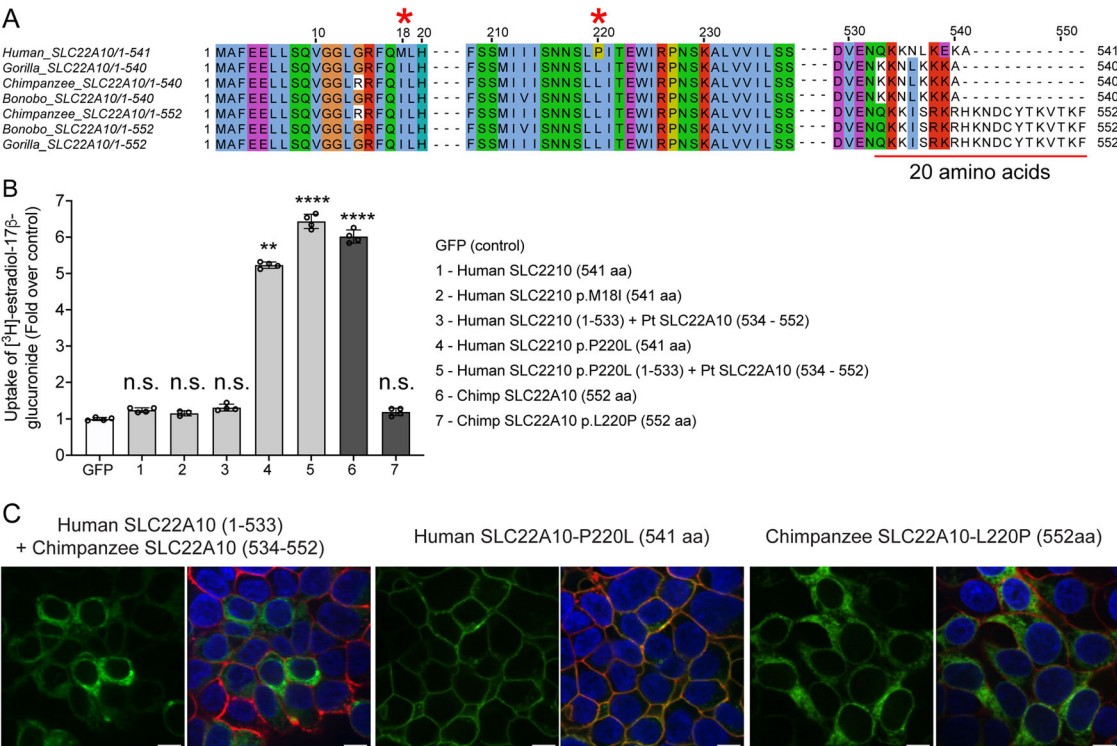

**Fig. 3 | A single mutation of proline to leucine at amino acid position 220 of human SLC22A10 significantly enhances the accumulation of [³H]-estradiol-17β-glucuronide in HEK293. A** The amino acid sequence alignment of human SLC22A10 and SLC22A10 from great apes (chimpanzee, bonobo and gorilla) shows that only the amino acids at positions 18 and 220 differ between the human ortholog and orthologs from great apes. Additionally, there are several amino acid differences starting at position 533. **B** The uptake of [³H]-estradiol-17β-glucuronide in HEK293 Flp-In cells transiently transfected with plasmids encoding human SLC22A10 with reference amino acids or amino acids that are similar to those found in other great apes, namely SLC22A10-p.M18I and SLC22A10-p.P220L. A chimeric protein consisting of the first 533 amino acids of human SLC22A10 and the last 19 amino acids of chimpanzee SLC22A10 (534-552) was also evaluated, but did not significantly accumulate [³H]-estradiol-17β-glucuronide compared to the chimeric protein with p.P220L. The fold uptake of the substrate, relative to the control (GFP) cells, was plotted based on one representative experiment conducted in triplicate wells (mean ± s.d.). The statistical significance for cells transfected with SLC22A10 #4 (Human SLC22I0 p.P220L (541 aa)), #5 (Human SLC22I0 p.P220L (1-533) + Pt

SLC22A10 (534 − 552)) and #6 (Chimp SLC22I0 (552 aa)) is p < 0.001. The scatter plot with bars show the fold uptake of [³H]-estradiol-17β-glucuronide relative to the negative control. HEK293 Flp-In cells transiently transfected with the GFP vector served as the negative control. The plot displays the mean +/− standard deviation of four technical replicates (*n* = 1 shown as a representative experiment). Source data are provided as a Source Data file. Statistical significance was determined using a one-way analysis of variance (ANOVA) with Dunnett's multiple comparison test. n.s.: non-significant, ****p value < 0.001. Similar results were obtained in three independent experiments. **C** This figure shows the plasma membrane localization of SLC22A10 conjugated to green fluorescent protein (GFP) in HEK293 Flp-In cells. The GFP tag is located at the N-terminus of SLC22A10. Confocal imaging revealed that human SLC22A10-p.P220L localizes primarily to the plasma membrane of the cell, while there was no localization to the plasma membrane in cells expressing a chimeric protein or chimpanzee SLC22A10 with proline at the 220 amino acid position. Blue: DNA stain Hoechst marks the cell nucleas; Red: Plasma membrane marker WGA; Green: SLC22A10. Yellow: Merge. The figure shows a representative image from two technical replicates. Scale bar: 10 μM.

in European (reported in the Single Nucleotide Polymorphism Database, dbSNP, version 156). Nonsense-mediated decay (NMD) is known to be triggered by premature stop codons. The Genotype-Tissue Expression (GTEx) project uncovered a significant association between the rs1790218 variant, which encodes the A-allele and p.Trp96Ter, and a considerable decrease in the transcript level of SLC22A10 samples (source: https://gtexportal.org/home/snp/rs1790218). This finding suggests that individuals carrying one copy of the p.Trp96Ter likely to exhibit an even lower protein signal in the liver in comparison to individuals that harbor p.Trp96.

### Mutagenesis of a single amino acid, at position 220 of the human SLC22A10 ortholog, to the respective amino acid in great apes rescues the function of human SLC22A10

The alignment of human SLC22A10 and primate ortholog sequences revealed differences in amino acid positions p.Met18Ile and p.Pro220Leu (Fig. 3A). Interestingly, site-directed mutagenesis experiments demonstrated that the substitution of proline with leucine at position 220 (p.Pro220Leu) restored the plasma membrane localization and function of human SLC22A10, but the substitution of

methionine with isoleucine at position 18 (p.Met18Ile) did not have any effect (see Fig. 3B). In contrast, the replacement of leucine with proline at position 220 (p.Leu220Pro) abolished the localization and function of the chimpanzee SLC22A10 (see Figs. 3B and 3C). Additionally, we observed that a human-chimpanzee chimera protein, consisting of a fusion of human SLC22A10 (1-533) with chimpanzee SLC22A10 (534-552), while retaining the proline residue at position 220, showed no function (Figs. 3B and 3C). However, with a leucine substitution at position 220, SLC22A10 remained functional (Fig. 3B).

### Human-specific p.Leu220Pro predates the common ancestor of modern humans and Neanderthals

Human-specific SLC22A10 variants likely emerged following the divergence of chimpanzee and human lineages. To better understand when SLC22A10 became non-functional during human evolution, we leveraged genomic data from archaic hominins[15–18] and ancient humans[19]. We first considered the fixed human variant at chr11: 63,064,927 (hg19), which results in the p.Leu220Pro amino acid in humans change compared to other apes. All four high-coverage archaic hominin genomes, representing three Neanderthals and a

**Table 1 | The protein levels of SLC22A10 in HEK293 Flp-In cells that were transfected with expression vectors containing open reading frames of SLC22A10 orthologs**

| HEK293 Flp-In Cells | [a]Human SLC22A10 protein (fmol/μg protein) | [b]Other SLC22A10 isoforms (see UniprotID below) (fmol/μg protein) |
|---|---|---|
| Flp-In only | 0.00 | 0.01 ± 0.014 |
| Chimpanzee SLC22A10 (552 amino acid) | 0.00 | 10.23 ± 4.23 |
| Chimpanzee SLC22A10-L220P (552 amino acid) | 0.00 | 2.1 ± 1.64 |
| Human SLC22A10 (541 amino acid) | 0.083 ± 0.052 | 1.59 ± 1.51 |
| Human SLC22A10-P220L (541 amino acid) | 1.00 ± 0.21 | 14.58 ± 8.99 |

The protein levels are reported in fmol/μg protein and the values are the mean ± standard deviation from two independent experiments, each with one technical replicate (see Source Data file the results from independent experiments). Source data are provided as a Source Data file. The proteomic analysis was conducted using UniProt IDs to identify and characterize the detected SLC22A10 protein in the sample. Specifically, the SLC22A10 protein levels were searched against the UniProt database to find matches with the SLC22A10 protein sequence.
[a]Protein ID: Q63ZE4.
[b]Uniprot Protein IDs for human and chimpanzee SLC22A10 protein sequence:
Human SLC22A10 (541 aa) - Q63ZE4.
Chimpanzee SLC22A10 (540 aa) - A0A2I3TWY5.
Chimpanzee SLC22A10 (552 aa) - A0A2J8LDY1.
Human SLC22A10 (205 aa) - E9PIT2.
Human SLC22A10 (264 aa) - E9PJB1.
Human SLC22A10 (372 aa) - E9PMM0.
Human SLC22A10 (155 aa) A0A804HHY1.

Denisovan, harbor a cytosine at this position, suggesting that this fixed difference evolved before the common ancestor of archaic hominins and modern humans, ~500 to 700 kya[15,20]. Next, we evaluated the p.Trp96Ter nonsense polymorphism at chr11: 63,057,925 (hg19). This variant was not assayed in archaic hominin genomes; however, we identified two SNPs in high linkage disequilibrium (LD) with the polymorphism that were also genotyped in ancient human samples. Variants in high LD constitute a haplotype and enable indirect dating of the nonsense variant. We detected both alleles in Eurasians as early as 30,000 years ago (Supplementary Fig. 2). Further, analyses of modern human genomes estimate polymorphism itself likely evolved ~ 120 kya (Supplementary Fig. 3). Third, we assessed the high-frequency insertion largely absent in other apes at chr11: 63,078,478 (hg19). While this variant could not be assessed using ancient DNA, we did not find LD between alleles at this locus and alleles at the p.Trp96Ter polymorphism indicating that these variants occur on separate haplotypes (Supplementary Table 1). This supports the hypothesis that the insertion emerged separately from the polymorphic stop and is also ancient; however, it is challenging to determine which arose first. Taken together, data from ancient DNA reveal that the p.Leu220Pro mutation predates the archaic hominin-modern human common ancestor and that both polymorphisms are also likely ancient.

**Human and chimpanzee SLC22A10 with Pro220 exhibit lower protein expression compared to orthologs with Leu220**

The objective of this study was to analyze the protein expression of SLC22A10 in HEK293 Flp-In cells that were transfected with either vector only, or the cDNA of human or chimpanzee SLC22A10. This was achieved by quantifying the global proteomes of the cells, with a specific focus on amino acids at position 220 of SLC22A10. Comparable transcript levels of SLC22A10 in HEK293 cells that over-expressed either human or chimpanzee SLC22A10, as well as the respective variants (p.P220L or p.L220P), were observed (Supplementary Fig. 4). However, as illustrated in Table 1, lower protein expression levels for the human SLC22A10 reference (Proline220) and chimpanzee SLC22A10-L220P were observed when compared to SLC22A10 with leucine at the 220 amino acid position. The results showed that protein levels of human SLC22A10 are approximately 10-fold lower in cells expressing human SLC22A10-Pro220 compared to human SLC22A10-Leu220 (Table 1) suggesting that human SLC22A10 is transcribed but the protein is unstable. The lower overall protein expression may explain the lack of detectable expression of human SLC22A10 on the plasma membrane in contrast to orthologs (both human and chimpanzee) of SLC22A10 with Leu220. In particular, human SLC22A10 was

not detected on the plasma membrane (Fig. 1B), whereas the mutant, SLC22A10-Leu220 exhibited expression on the plasma membrane (Fig. 3C).

**Human SLC22A10 with Pro220 is predicted to have poor stability compared to orthologs with Leu220**

We tested the effect of the P220L mutation on protein stability using a prediction pipeline tuned for transmembrane proteins[21]. The Rosetta physics-based score suggested that P220L is a stabilizing mutation with a ΔΔG value of −9.84 Rosetta Energy Units (REU). Using rank-normalized evolutionary-based ΔΔE scores, we calculated values of 0.0 for leucine and 0.16 for proline at this position. This combination of a higher ΔΔE value and a less negative ΔΔG value suggests that the Pro220 creates a potential loss of function protein, likely due to decreased stability and lower cellular abundance[22]. To test this idea, we explored the relative stability of human SLC22A10 (Pro220) and SLC22A10 (Leu220) using flow cytometry. Specifically, HEK293 Flp-In cells were stably transfected with vector control, human SLC22A10 reference (Pro220), or human SLC22A10 mutant (Leu220). In these cells, we measured both the stable level of GFP fluorescence and the response of that fluorescence to treatment with 30 nM of the 26 S proteosome inhibitor, bortezomib. For the L220 mutant, GFP fluorescence is high in comparison to the empty vector control and increases minimally upon treatment with bortezomib (Supplementary Fig. 5), suggesting that this protein is relatively stable and not actively degraded by the proteosome. In contrast, the GFP fluorescence of SLC22A10 reference (Pro220) occupies a value midway between that of the empty vector control and the SLC22A10 mutant (Leu220), confirming that it is less stable. Upon bortezomib treatment, the GFP fluorescence of SLC22A10 reference (Pro220) increases, such that it is comparable to that of SLC22A10 mutant (Leu220) (Supplementary Fig. 5). Thus, SLC22A10 (Pro220) appears to be relatively unstable and degraded by the proteosome, such that its protein levels can be partially restored by proteasome inhibition.

**Analyses of SLC22A10 isoforms in chimpanzee, bonobo, orangutan and gibbon**

Upon examining various databases (UniProt, Ensembl, and NCBI nucleotide) that predict SLC22A10 isoforms, we observed that chimpanzee, bonobo, orangutan, and gibbon are predicted to have different isoforms, including (i) chimpanzee (XM_024347215.1, 533 amino acid); (ii) bonobo (XM_034932815.1, 538 amino acid); (iii) gibbon (XM_003274123.1, 533 amino acid); and (iv) orangutan (XM_024255628.1, 533 amino acid). The shorter isoforms are derived

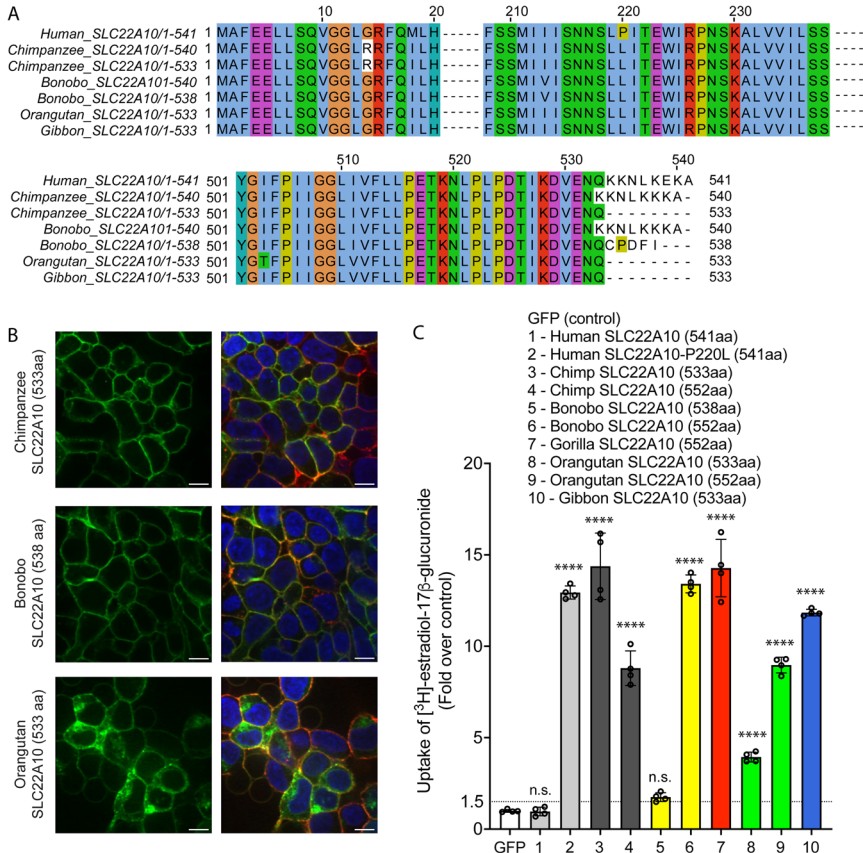

**Fig. 4 | SLC22A10 of chimpanzees, bonobos, orangutans, and gibbons are predicted to have shorter isoforms expressing 533 or 538 amino acids. A** A comparison of the SLC22A10 amino acid sequence of humans, chimpanzees, bonobos, and orangutans, which express 533 (chimpanzee, orangutan, gibbon), 538 (bonobo), 540 (bonobo, chimpanzee), or 541 (human) amino acids, shows that the major differences are at the end of the SLC22A10 sequence. **B** Confocal imaging revealed that SLC22A10 from chimpanzees and bonobos (isoforms expressing 533 or 538 amino acids) primarily localize to the plasma membrane of the cell, whereas weaker localization was observed for orangutan SLC22A10 (533 amino acids) to the plasma membrane of the cell. GFP conjugated to SLC22A10 was used for this experiment. Blue: DNA stain Hoechst marks the cell nucleas; Red: Plasma membrane marker WGA; Green: SLC22A10. Yellow: Merge. The figure shows a representative image from two technical replicates. Scale bar: 10 μM. **C** The uptake of [³H]-estradiol-17β-glucuronide in HEK293 cells was observed after transient

transfection of plasmids encoding human SLC22A10 with reference amino acids or SLC22A10 with reference amino acids of other great apes with different isoforms. The results showed that SLC22A10 isoforms expressing 533 and 552 amino acids significantly accumulate the substrate. However, no significant accumulation was observed in cells transfected with the bonobo SLC22A10 isoform expressing 538 amino acids. The scatter plot with bars shows the fold uptake of [³H]-estradiol-17β-glucuronide relative to the negative control. HEK293 Flp-In cells transiently transfected with the GFP vector served as the negative control. The plot displays the mean +/- standard deviation of four technical replicates ($n=1$ shown as a representative experiment). Source data are provided as a Source Data file. Statistical significance was determined using a one-way analysis of variance (ANOVA) with Dunnett's multiple comparison test. n.s.: non-significant, ****$p$ value < 0.001. Similar results were obtained in two independent experiments.

from alternative acceptor sites (chimpanzee and bonobo) and exon extensions (orangutan and gibbon) resulting in 533-amino acid proteins (see Fig. 4A, Supplementary Fig. 6). We conducted experiments using HEK293 Flp-In cells that were stably transfected with GFP-conjugated chimpanzee, bonobo, and gorilla SLC22A10 to examine the localization and function of the shorter isoforms. The HEK293 Flp-In cells that were stably transfected with GFP-conjugated chimpanzee SLC22A10, consisting of 533 amino acids, showed a clear plasma membrane localization. However, it appears that the bonobo SLC22A10, which consists of 538 amino acids, exhibited weaker localization, and the orangutan SLC22A10, also comprising 533 amino acids, showed mixed localization (Fig. 4B). Our results indicated that the SLC22A10 533 amino acid isoform from chimpanzee, orangutan and gibbon accumulated [³H]-estradiol-17β-glucuronide and [³H]-folic acid, but not [³H]-estrone sulfate and [³H]-taurocholic acid, similar to the SLC22A10 isoform with 552 amino acids (Fig. 4C, Supplementary Fig. 7). Moreover, we observed that longer isoforms of SLC22A10 created for the bonobo and orangutan, which were not predicted to express these isoforms (552 amino acids), accumulated [³H]-estradiol-

17β-glucuronide, and were also expressed on the plasma membrane (Supplementary Fig. 7). However, the uptake by bonobo SLC22A10, comprising 538 amino acids, is not significant, and there is a weaker fold uptake by orangutan SLC22A10, consisting of 533 amino acids, consistent with its mixed localization on the plasma membrane. (Fig. 4B and C). During the revision of this article, it was noted that the NCBI updated the sequence of bonobo SLC22A10 from 538 amino acids to 533 amino acids. This sequence of bonobo SLC22A10, comprising 533 amino acids, exhibited a significant uptake of [³H]-estradiol-17β-glucuronide and demonstrated clear plasma localization (Supplementary Fig. 8).

## Uptake of various steroid glucuronides by chimpanzee SLC22A10 reveals that 17β-glucuronides are preferred over 3α-glucuronide conjugates

Our experiments revealed significant accumulation of estradiol-3β-glucuronide, estradiol-17β-glucuronide, estrone-3β-glucuronide and testosterone-17β-glucuronide (Fig. 5A). We also observed weak but significant accumulation of androstanediol-3α-glucuronide, but no

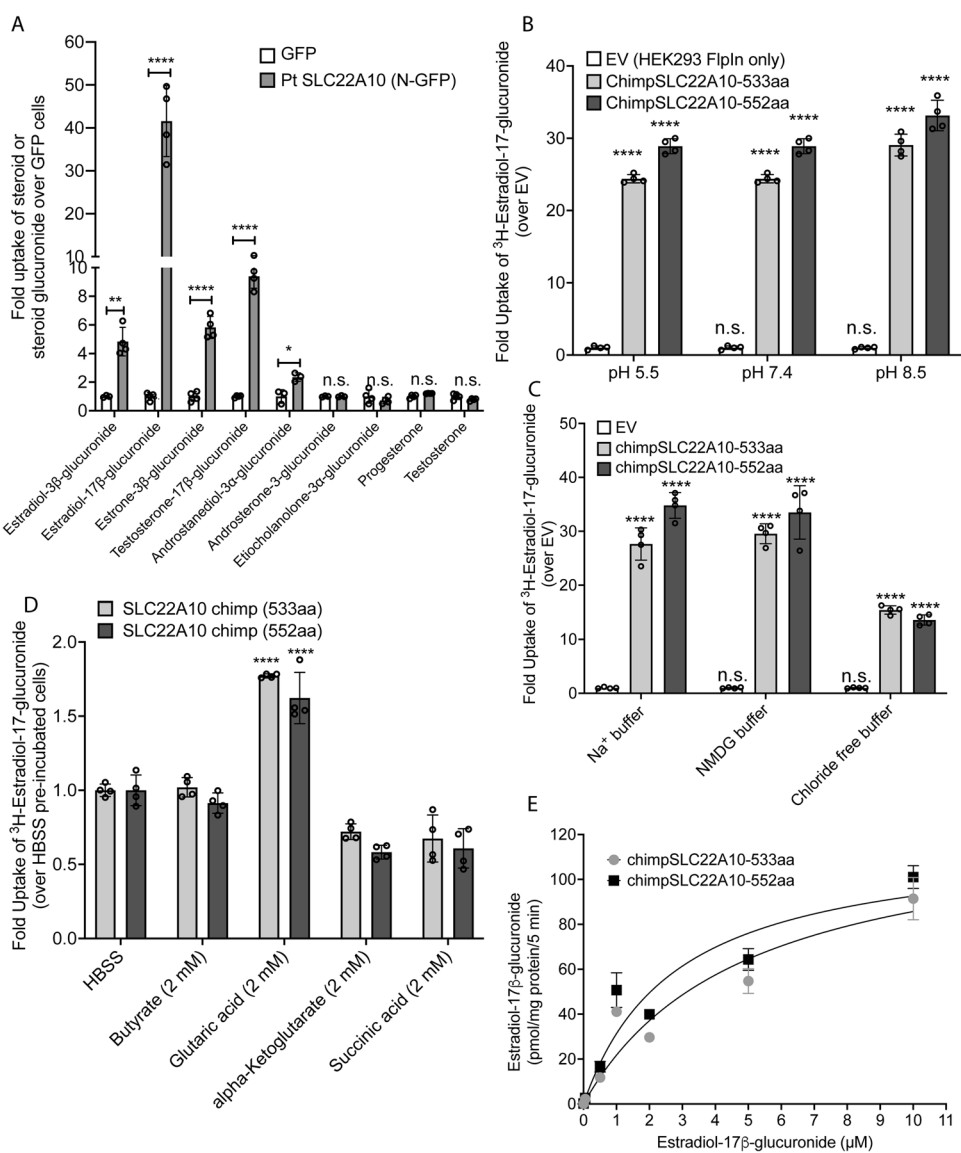

significant accumulation of androsterone-3-glucuronide, etiocholanolone-3α-glucuronide, progesterone, and testosterone (Fig. 5A). Although the data are limited, it is intriguing to note the substrate preference of SLC22A10. Specifically, 17β-glucuronides appear to be favored over 3α-glucuronide conjugates.

### Chimpanzee SLC22A10-mediated uptake is sodium independent, pH independent, and trans-stimulated by glutaric acid

Transporters in the SLC22 family may be secondary active, in which case they rely on various sources of energy to mediate active flux of their substrates. Accordingly, we examined the transport mechanism for two isoforms (533 and 552 amino acids) of chimpanzee SLC22A10. At the three pH levels evaluated, the uptake of [³H]-estradiol-17β-glucuronide by both isoforms of chimpanzee SLC22A10 were similar (Fig. 5B). While the uptake was not dependent on sodium (Fig. 5C), it was partially reduced in the absence of chloride in the buffer (Fig. 5C). Several organic anion transporters in the SLC22 family are *trans*-stimulated by dicarboxylic acids[3,23]. Similarly, we observed that the uptake of [³H]-estradiol-17β-glucuronide was *trans*-stimulated by glutaric acid but not significantly by other dicarboxylates (α-ketoglutarate and succinic acid) or the monocarboxylic acid, butyrate (Fig. 5D). The uptake kinetics of [³H]-estradiol-17β-glucuronide exhibited saturable characteristics with virtually identical Km values for the two protein

isoforms at $3.28 \pm 2.25\,\mu M$ and $2.16 \pm 0.59\,\mu M$ for the 533 aa and 552 aa isoforms, respectively (Fig. 5E).

## Discussion

Transporters play a major role in total body homeostasis as they function to regulate the levels of many solutes including endogenous metabolites, essential nutrients, and environmental toxins. Over 30% of genes in the SLC superfamily have no known function[1]. Identifying the substrates or deorphaning transporters presents significant challenges due to their diverse structural and functional characteristics, as well as the intricate cellular and physiological context in which they operate. Overcoming these challenges necessitates the implementation of innovative approaches that integrate computational predictions, high-throughput screening, functional assays, and targeted experimental investigations[1,24]. Recent advancements in transporter research in the past four years have led to the identification of ligands for several transporters in the organic ion transporter family, SLC22. Notably, SLC22A14, SLC22A15, and SLC22A24 have been successfully characterized[2,3,25]. In this study, we pursued a distinct approach to deorphaning a human SLC22 family member, SLC22A10 by investigating the function of great ape orthologs. Our comprehensive investigations have yielded five major findings that contribute to our understanding of the function and evolution of SLC22A10 in higher order primates.

**Fig. 5 | This figure presents information about the transport mechanism and kinetics of chimpanzee SLC22A10. A** The uptake of seven steroid glucuronides and two steroids in HEK293 Flp-In cells stably transfected with chimpanzee SLC22A10 (552 amino acids) was measured using LC-MS/MS to determine the accumulation of the compounds. The scatter plot with bars shows the fold uptake of [³H]-estradiol-17β-glucuronide relative to the negative control. HEK293 Flp-In cells stably transfected with the GFP vector served as the negative control. The plot displays the mean +/- standard deviation of four technical replicates ($n = 1$ shown as a representative experiment). Source data are provided as a Source Data file. Two-tailed unpaired *t*-tests were performed to compare the two groups (GFP vs. chimpanzee SLC22A10-GFP tagged). n.s.: non-significant, *$p$ value < 0.05, **$p$ value < 0.01, ****$p$ value < 0.001. **B** The effect of pH on accumulation of [³H]-estradiol-17β-glucuronide in HEK293 Flp-In cells stably transfected with chimpanzee SLC22A10 isoforms expressing 533 and 552 amino acids was investigated. For each pH condition, the scatter plot with bars shows the fold uptake of [³H]-estradiol-17β-glucuronide relative to the negative control. HEK293 Flp-In cells stably transfected with the empty vector (EV) served as the negative control. The plot displays the mean +/- standard deviation of four technical replicates ($n = 1$ shown as a representative experiment). Source data are provided as a Source Data file. For each pH condition, statistical significance was determined using a one-way analysis of variance (ANOVA) with Dunnett's multiple comparison test. n.s.: non-significant, ****$p$ value < 0.001. **C** The effect of sodium and chloride on accumulation of [³H]-estradiol-17β-glucuronide in HEK293 Flp-In cells stably transfected with 533 and 552 amino acid isoforms of chimpanzee SLC22A10 was investigated. For each buffer condition, the scatter plot with bars shows the fold uptake of [³H]-estradiol-17β-glucuronide relative to the negative control. HEK293 Flp-In cells stably transfected with the empty vector (EV) served as the negative control. The plot displays the mean +/− standard deviation of four technical replicates ($n = 1$ shown as a

representative experiment). Source data are provided as a Source Data file. For each buffer condition, statistical significance was determined using a one-way analysis of variance (ANOVA) with Dunnett's multiple comparison test. n.s.: non-significant, ****$p$ value < 0.001. **D** The effects of *trans*-stimulation of [³H]-estradiol-17β-glucuronide uptake by chimpanzee SLC22A10 was determined. Uptake was *trans*-stimulated by preloading the cells with 2 mM of butyrate, glutaric acid, alpha-ketoglutarate, or succinic acid for 2 hours, and then measuring the uptake of [³H]-estradiol-17β-glucuronide after 15 minutes. The data are presented as mean ± S.D. and were normalized by setting the uptake of SLC22A10-expressing cells trans-stimulated by HBSS to 1.0. *Trans*-stimulation of [³H]-estradiol-17β-glucuronide by glutaric acid was observed for both isoforms of chimpanzee SLC22A10. The scatter plot with bars shows the fold uptake of [³H]-estradiol-17β-glucuronide relative to either chimpanzee SLC22A10 (533aa) or SLC22A10 (552aa) treated with HBSS for 2 hours. Each of this isoform treated with HBSS for 2 hours served as the negative control. The plot displays the mean +/- standard deviation of four technical replicates ($n = 1$ shown as a representative experiment). Source data are provided as a Source Data file. For each chimpanzee SLC22A10 isoform, statistical significance was determined using a one-way analysis of variance (ANOVA) with Dunnett's multiple comparison test. n.s.: non-significant, ****$p$ value < 0.001. **E** The kinetics of [³H]-estradiol-17β-glucuronide uptake for chimpanzee SLC22A10 isoforms expressing 533 and 552 amino acids were analyzed. The uptake rate was evaluated at 5 minutes and the data were fit to a Michaelis-Menten equation. To fit the kinetic curve to a Michaelis-Menten equation, the concentration of estradiol-17β-glucuronide is set up to 10 μM. The plot displays the mean +/- standard deviation of three technical replicates ($n = 1$ shown as a representative experiment). Source data are provided as a Source Data file. The figure shows a representative plot from one experiment. Similar results were obtained in two independent experiments (**A** to **E**).

First, SLC22A10 functions as a steroid glucuronide transporter in great apes (as shown in Figs. 1 and 2). Unlike other organic anion transporters in the SLC22 family such as SLC22A6, SLC22A7, and SLC22A8, which transport a variety of organic anions including uric acid, steroid glucuronides, bile acids, steroid sulfates, dicarboxylic acids, and phosphate-containing nucleotides[6], the ortholog of SLC22A10 from primates (chimpanzee, bonobo, gorilla, orangutan and gibbon) transported primarily estradiol-17β-glucuronide (Supplementary Fig. 1) with significantly weaker uptake of other organic anions such as folic acid and methotrexate (Supplementary Figs. 1 and 7). Interestingly, the SLC22A10 ortholog in the squirrel monkey, a New World monkey, preferred estrone sulfate over estradiol-17β-glucuronide (Supplementary Fig. 7) whereas the gene encoding SLC22A10 is absent in Old World monkeys (*Cercopithecidae* family) (ensemble release 110). These results suggest an evolving role of the function of the transporter in non-human primates. In both chimpanzee and squirrel monkey, SLC22A10 is expressed specifically in the liver (NHPRTR, http://nhprtr.org/), consistent with a role in steroid metabolism. The closest paralog of SLC22A10 is SLC22A24, a recently deorphaned transporter[3]. Both transporters take up steroid conjugates though chimpanzee SLC22A10 has a narrower substrate specificity than the human SLC22A24. In contrast, SLC22A8, a multispecific transporter, expressed primarily in the kidney, for an array of drugs, toxins, and other metabolites, is not typically associated with sex steroid metabolism due to its capacity to transport a diverse set of molecules with varying affinities[26,27]. In *Slc22a8* null mice, a spectrum of non-sex steroid conjugates was shown to be significantly elevated in the plasma[27–29]. In addition, mouse *Slc22a20* (Oat6), which is expressed in Sertoli cells of the testis, is known to transport steroid conjugates (sulfates) and is inhibited by various canonical anions (probenecid, ochratoxin A, p-aminohippurate)[30] as well as odorant organic anions (e.g., 4-bromobutyric acid, propanoic acid, hexanoic acid, 2-methylbutyrate)[31,32]. Phylogenetic tree analysis conducted by other research groups[33,34] and Fig. 1A from our study shows that SLC22A8 and Slc22a20 are distant from SLC22A10 within the OAT clade. A recently published study showed that other members of the mouse OAT clade (Slc22a19, Slc22a26, Slc22a27, Slc22a28, Slc22a29,

Slc22a30), which have no direct orthologs in humans within the SLC22 family, transport thyroid hormone and thyroid hormone sulfates[33], and these genes are further away from human SLC22A10 on the phylogenetic tree (Fig. 1A).

Our second major finding is that the function of SLC22A10 is lost in humans. That is, while human SLC22A10 is transcribed, the protein expression is undetectable in human cell lines transfected with human SLC22A10 and in human liver tissue[35–39] (https://www.proteinatlas.org/ENSG00000184999-SLC22A10/tissue), consistent with a loss of function of the gene in humans. Indeed, proteomic analysis demonstrated that the reference proline (Pro220) displayed significantly lower protein expression compared to the mutated form, Leu220 (see Table 1). Further, as Pro220 is fixed in humans resulting in a loss of function, the gene has been under less selective pressure. SLC22A10 contains a common nonsense variant, p.Trp96Ter (rs1790218), with high allele frequencies in human populations (available from dbSNP). This variant, associated with the A-allele, triggers nonsense-mediated decay (NMD), leading to significantly reduced SLC22A10 transcript levels in samples (source: https://gtexportal.org/home/snp/rs1790218, GTEx project). This finding is relevant to how the protein itself (though not a functioning protein) is being lost. During our efforts to clone the human SLC22A10 gene using pooled human liver samples, we selected 27 individual colonies for further analysis and subsequent sequencing. Only four of these colonies contained the complete transcript spanning 1626 base pairs. The remaining 23 out of 27 colonies selected had a identical 219 bp deletion, resulting in a truncated sequence of 1407 bp, corresponding to 469 amino acids (Supplementary Fig. 9). However, NCBI predicted a different deletion of 324 bp, starting at the same GT splice donor site that we observed, but extending to a further GT acceptor site (NCBI Reference Sequence: XM_047426921).

Our third major finding is that Leu220Pro alone caused the loss of function of the gene in hominins. That is, a single proline substitution resulted in no expression of SLC22A10 on the plasma membrane and significantly reduced protein abundance (Table 1). When Pro220 in human SLC22A10 was mutated to Leu220, it acquired the functional capacity observed in chimpanzee SLC22A10, resulting in substantial accumulation of the substrate, estradiol-17β-glucuronide. Conversely,

when the amino acid at position 220 in chimpanzee SLC22A10 was mutated to proline, the uptake of estradiol-17β-glucuronide was completely abolished (Fig. 3B). Cordes F. et al. (2002) reported that distortions of transmembrane helices can be induced by the presence of proline[40]. Using AlphaFold2 model of human SLC22A10, we observed that the proline 220 in humans introduces a kink in the alpha-helix, which in turn can affect the conformation of the 225-230 loop and. In addition, based on the ΔΔG calculation the proline 220 is significantly less stable with a less negative value of ΔΔG. Treatment with the proteasome inhibitor bortezomib led to an increased GFP fluorescence in HEK293 cells expressing GFP-tagged SLC22A10 reference (P220), but not in HEK293 cells expressing the GFP-tagged SLC22A10 mutant (L220). Together, these data show that human SLC22A10 undergoes degradation in the proteasome due to its inherent instability (Supplementary Fig. 5). Based on extensive genomics data-sets from large-scale populations such as gnomAD[41], GDBIG (http://www.bigcs.com.cn/), TopMed Freeze 8[42], and GME Variome[43] and the All of Us Research Hub Data Browser[44], we found two individuals who carry the SLC22A10-Leu220 mutation (heterozygous) in the All of Us Research Hub, as well as two more in gnomAD (v4.0)[45]. Additionally, archaic hominin genomes share the fixed p.Leu220Pro difference with modern humans, suggesting that this mutation emerged following Pan-Homo divergence and before modern humans diverged from archaic hominins. Further, the SLC22A10-Trp96Ter variant appears to be ~ 120 kya in age.

Our fourth finding unveils the transporter activity and expression of the predicted isoforms of SLC22A10 on the plasma membrane in great apes (refer to Fig. 2, Fig. 3, and Fig. 4). Importantly, among the great apes, three isoforms were initially predicted: 533, 540, and 552 amino acids in length. However, further investigation revealed that only the isoforms with 533 and 552 amino acids were actually expressed on the plasma membrane and exhibited transporter activity, as depicted in Fig. 4 and Fig. 5. In Fig. 2C, the last exons (exon number 10) for humans and other great apes is presented. It is worth noting that human SLC22A10 is predicted to consist of 541 amino acids. The gene annotation of chimpanzees and other primates in Ensembl is limited by an anthropocentric approach that heavily relies on human annotation as a reference. In the current version 110 of Ensembl.org, the chimpanzee, gorilla, bonobo gene are annotated with 540 amino acids. This annotation utilizes a non-canonical splice site at the end of exon 9 ('CA') instead of the canonical donor splice site ('GT'), which is not supported by our chimpanzee liver RNA-seq data[13] (Supplementary Fig. 6 and Fig. 2C). In contrast, NCBI predicted that chimpanzee possess isoforms with 533 and 552 amino acids, which is consistent with our observation from chimpanzee liver RNAseq data[13]. Within the SLC22A family, several transporters, including SLC22A7[46] and SLC22A24[3] exhibit distinct isoforms.

Our fifth finding suggests that SLC22A10 plays a significant role in the liver, and provides some insight into the biological role of the transporter. RNA-seq data indicate that the SLC22A10 gene is specifically expressed in the liver in chimpanzees, bonobos, and gorillas, with very low expression in the brain[47], mirroring the transcriptomic data in the human GTEx project. The liver is a major organ for steroid hormone metabolism, such as UDP-glucuronosyltransferases (UGT), which forms steroid glucuronide conjugates to enhance water-solubility for excretion by the kidney[48]. Similarly, the liver is also abundant in beta-glucuronidase[49], which removes glucuronides and forms the parent compound, such as steroids. Notably, several steroid conjugate influx and efflux transporters are expressed in the liver, some of which have greater specificity for steroid conjugates and many other substrates (e.g., *SLCO1B1*, *SLCO1B3*), while others have narrow specificities (e.g., *SLC22A9*, known to transport sulfate conjugates[33,50]). SLC22A10 in great apes exhibited significant uptake of steroid glucuronides (Fig. 5, Supplementary Fig. 1) but not other well-known anions, which are substrates of other SLC22 family members, such as bile acids and dicarboxylic acids

(Fig. 1, Supplementary Fig. 1, Supplementary Fig. 7). To our knowledge, SLC22A10 is the only specific steroid glucuronide transporter in the liver, possibly due to its specialized role in the uptake of steroid conjugates in the liver to expose them to beta-glucuronidase, forming the parent steroid. This recycling process from glucuronide metabolites to the parent compound (in this case steroids) is a common process in biology, and occurs for bile acids, bilirubin and many clinically used drugs. Although the transporter is primarily expressed in liver, it appears to be expressed at lower levels in neurons (https://www.proteinatlas.org/ENSG00000184999-SLC22A10/single+cell+type).

Overall, the broader impact of this study on SLC22A10, an orphan transporter, is multifaceted and significant. The study not only sheds light on the specific characteristics of SLC22A10 but also contributes to a broader understanding of genetic evolution, the role of orphan transporters in biological processes and potential applications in medicine and pharmacology. Our studies revealed that the activity of SLC22A10 has evolved in primates, with Old World monkeys lacking SLC22A10 orthologs, and New World monkeys exhibiting a different substrate preference compared to great apes. In addition, SLC22A10 was rendered nonfunctional in humans by a single missense mutation during hominin evolution after our shared ancestor with the chimpanzee. This missense mutation resulted in a complete loss of human SLC22A10 transporter activity, due to lack of protein expression on plasma membrane and reduce protein abundance. With time, the gene has accumulated additional mutations, including a stop codon (p.Trp96Ter), which has led to reductions in the levels of the mRNA transcript and corresponding reductions in protein level. This finding highlights the complex nature of evolutionary processes, specifically how genetic mutations can render certain genes nonfunctional in some species while remaining active in others. Human SLC22A10 gene exhibits features that classify it as unitary pseudogene[51,52]. A pseudo-gene can be defined by the loss of the original function due to errors during transcription or translation, or as a gene producing a protein that does not have the same functional repertoire as the original gene[52]. Consequently, a pseudogene will not necessarily evolve under a neutral theory of molecular evolution. Pseudogenes can be categorized into 3 different types depending on their functional state. These include exapted pseudogenes, which have gained a new biological function; "dying" pseudogenes, which still have some transcriptional activity; and "dead" pseudogenes, which do not exhibit any signs of activity and evolve under the neutral theory[51]. Based on the evidence at hand, we cannot differentiate if the pseudogene is exapted or dying. However, the SLC22A10 gene is a well-established gene that originated from the last common ancestor of boreoeutheria, with no functioning counterparts in the human genome. Thus, we can classify human SLC22A10 an unitary pseudogene. Examples of unitary pseudogene in human is MUP (major urinary protein), whereas uricase (UOX) and GULO (L-Gulonolactone oxidase) are well established unitary pseudogenes that were inactivated before the separation of human and chimpanzee[53], There are known pseudogenes in the human SLC superfamily, for example, SLC22A20, functions as a steroid conjugate transporter (esterone sulfate) in mice and is expressed in the blood-testes barrier[54]. Future studies are needed to determine whether the loss of function human SLC22A10-P220 is a favorable situation for humans and whether similar mechanisms have led to the inactivation of other orphan genes in the human genome. In addition, studies are needed to determine the role of SLC22A10 in biological processes such as the transport of sex steroid conjugates in great apes.

## Methods

### Generation of SLC22A10 ortholog cDNA constructs

Sequences of SLC22A10 orthologs from various species, along with their respective sequence IDs, were obtained from ensembl.org (release version 104)[55] and the National Center for Biotechnology Information (NCBI). The gene's cDNA was synthesized using GenScript

Gene Synthesis and DNA Synthesis Services and subsequently inserted into the multiple cloning sites on the expression vectors pcDNA3.1(+) or pcDNA3.1( + )N-eGFP. The sequence of the constructs was verified through sequencing conducted by GenScript.

Human SLC22A10, NM_001039752, NP_001034841.3 (541 aa)
Chimpanzee SLC22A10, XM_016921102.2, XP_016776591.1 (552 aa)
Chimpanzee SLC22A10, ENSPTRT00000087785.1, A0A2I3TWY5 (540 aa)
Chimpanzee SLC22A10, XM_024347215.1, XP_024202983 (533 aa)
Bonobo SLC22A10, XM_055094430.1, XP_054950405.1 (533 aa)
Bonobo SLC22A10, XM_034932815.1, XP_034788706 (538 aa)
Bonobo SLC22A10, ENSPPAT00000058172.1, A0A2R9CA90 (540 aa)
Gorilla SLC22A10, XM_019037494.1, XP_018893039 (552 aa)
Gorilla SLC22A10, ENSGGOT00000014974.3, G3RG24 (540 aa)
Orangutan SLC22A10, XM_024255628.1, XP_024111396.1 (533 aa)
Orangutan SLC22A10, XM_054439153.1, XP_054295128.1 (552 aa)
Gibbon SLC22A10, XM_003274123.2, XP_003274171.2 (533 aa)
Squirrel monkey SLC22A10, ENSSBOT00000021267.1, A0A2K6 SBP5 (552 aa)

## Site-directed mutagenesis

Site-directed mutagenesis to create Methionine18 to Isoleucine18 in human SLC22A10, Proline220 to Leucine220 or vice versa in chimpanzee SLC22A10 and human SLC22A10, were performed using Q5 Site-Directed Mutagenesis Protocol from NEB (#E0554). NEBaseChanger, https://nebasechanger.neb.com/, is used to assist in the design of primers for the site-directed mutagenesis experiment. The following primers were ordered from IDT™ (Integrated DNA Technologies).

Primers for p.M18I:
mut_hsA10_c54C_F: GATTTCAGATcCTTCATCTGGTTTTTATTCTTC
mut_hsA10_c54C_R: TCCCAAGGCCTCCAACTT
Primers for p.P220L:
mut_hsA10_c659T_F: AATTCTTTGCtCATTACTGAG
mut_hsA10_c659T_R: ATTTGATATAATGATCATGGAAG
Primers for p.L220P:
mut_ptA10_c659C_F: AATTCTTTGCcCATTACTGAGTG
mut_ptA10_c659C_R: ATTTGATATAATGATCATGGAAG
Primers for deletion of A nucleotide in position c1607 and then inserting the tail at the last exon for human to chimpanzee (consist of 34 bp):
mut_hsA10_c1607Adel_F: TCTCAAGGAAAAGGCATAAATG
mut_hsA10_c1607Adel_R: TTTTTTTTTGATTTTCCACATCC
mut_hsA10_c1622ins_F: caaaagtgaccaaatttTAACTCGAGTCTAGAGGG
mut_hsA10_c1622ins_R: tgtagcaatcatttttaTGCCTTTTCCTTGAGATTTTTTTTTG

## Cell culture

Human embryonic kidney cell lines (HEK293) containing a Flp-In expression vector (HEK293 Flp-In) obtained from ThermoFisher Scientific (#R75007) were utilized in all experiments in this study. These cells have previously been employed for the generation of stable cell lines for transporter assays[2,3,23,46]. HEK293 Flp-In cells have not been authenticated. The HEK293 Flp-In cells were cultured in DMEM, high glucose (#11965118, ThermoFisher Scientific) supplemented with 10% fetal bovine serum (heat inactivate, #10438026, ThermoFisher Scientific). Penicillin-Streptomycin (#15070063, ThermoFisher Scientific) was added to DMEM media (50 unit/500 mL DMEM). During transfection and when cells were plated for transporter studies, media without penicillin/streptomycin supplementation were used. The cells were periodically screened for mycoplasma contamination (MycoProbe Mycoplasma Detection Kit, #CUL001B, Fisher).

## Generation of cells transiently or stably expressing cDNAs

Expression vectors of SLC22A10 orthologs were introduced into HEK293 Flp-In cells either through transient transfection or stable transfection using Lipofectamine LTX (Thermo Fisher Scientific). For transfections in a 48-well plate (seeding density: $1.0 \times 10^5$ cells/well), 200 ng of DNA and 0.4 µL of Lipofectamine LTX were utilized, while for transfections in a 100 mm tissue culture plate (seeding density: $4 \times 10^6$ cells/well), 10 µg of DNA and 44 µL of Lipofectamine LTX were used. More comprehensive methods for generating transiently or stably transfected cells have been described in our previous work (see reference[2,3]). In the case of transient transfection, cells were used for transporter studies (refer to the section titled "Transporter uptake studies") after 36-48 hours or for protein quantification after 72 hours. To establish stable cell lines, 3000 ng of DNA (SLC22A10 ortholog expression vectors) and 10.5 µL of Lipofectamine LTX were employed to transfect HEK293 Flp-In cells seeded in a 6-well plate (seeding density: $7-8 \times 10^5$ cells/well). After 48 hours, cells were transferred to a new 100 mm tissue culture plate and treated with 800 µg/mL Geneticin. Fresh media containing 800 µg/mL Geneticin was replenished every other day for 1 week. Stable cell lines were utilized for confocal imaging to determine the plasma membrane localization of SLC22A10 orthologs and their various isoforms. Unless specified otherwise, stable cell lines were used for transporter assays.

## Fluorescence microscopy

For the immunostaining experiments, HEK293 Flp-In stable cell lines expressing different SLC22A10 orthologs were cultured on poly-D-lysine-treated 12-well plates with sterile coverslips at a density of 200,000 cells per well. After two days of seeding when the cells reached 90-100% confluency, the staining procedure was conducted. On the day of staining, the cell culture media was carefully removed, and the cells were washed using cold Hank's Balanced Salt Solution (HBSS, #14025092, ThermoFisher Scientific). To initiate the staining process, the plasma membrane was first labeled using Wheat Germ Agglutin (WGA) Alexa Fluor 647 conjugate (Invitrogen Life Sciences Corporation) diluted at a ratio of 1:500 in HBSS, followed by a 15-minute incubation at room temperature. Following the staining step, the WGA solution was aspirated, and the cells were washed three times with HBSS. Subsequently, the cells were fixed with a solution of 3.7% formaldehyde in HBSS for 20 minutes. After the fixation step, the cells were washed three times with HBSS. To stain the nucleus, Hoechst solution (ThermoFisher Scientific Inc.) diluted at a ratio of 1:2000 in HBSS was applied to the cells and incubated for 20 minutes at RT, in darkness. After the staining period, the Hoechst solution was aspirated, and the cells were washed twice with HBSS. The coverslips were carefully mounted on Superfrost Plus Microscope Slides (ThermoFisher Scientific) using a small amount of SlowFade™ Gold Antifade Mountant (#S36940, ThermoFisher Scientific). The mounted slides were left to dry overnight in darkness before being imaged using an inverted Nikon Ti microscope equipped with a CSU-22 spinning disk confocal system available at the Center for Advanced Light Microscopy (CALM) at University of California San Franciso. The image acquisition settings were as follows: DAPI channel with a 300 ms exposure time and 50% laser power, FITC channel with a 300 ms exposure time and 25% laser power, and CY5 channel with a 100 ms exposure time and 5% laser power. Image alignment and merging were performed using Fiji software. This experimental protocol has been previously utilized and described in our published work[56,57].

## Transporter uptake studies

HEK293 Flp-In cells expressing SLC22A10 were seeded at a density of 120,000 to 150,000 cells/0.3 mL in poly-D-lysine-coated 48-well plates approximately 16 to 24 hours prior to conducting uptake studies. The uptake studies for transporters, described below, are methods published by our research group[2,3]. For transiently expressing SLC22A10,

the methods outlined in the previous section pertaining to transient expression in HEK293 Flp-In cells were followed prior to this step. Prior to uptake studies, the culture medium (Dulbecco's modified Eagle's medium, DMEM) supplemented with 10% fetal bovine serum was aspirated, and the cells were incubated in 0.8 mL of Hank's Balanced Salt Solution (HBSS) at 37 °C for 10-20 minutes. For screening radiolabeled compounds as SLC22A10 substrates, minute quantities of radiolabeled compounds ($^3$H or $^{14}$C) were diluted in HBSS (at ratios of 1:2000 or 1:3000) for uptake experiments. Unlabeled compounds were added to obtain specific concentrations, which are described in the Results section or figure legends along with the uptake times. Uptake reactions were terminated by washing the cells twice with 0.8 mL of HBSS buffer, followed by incubation in 750 μL of lysis buffer (0.1 N NaOH, 0.1% v/v SDS). A 690 μL portion of the cell lysate was transferred to scintillation fluid for scintillation counting. For pH dependence experiments, the HBSS buffer was adjusted to different pH levels (5.5, 7.4, and 8.5) using hydrochloric acid or sodium hydroxide. For sodium and chloride dependence studies, three distinct uptake buffers were employed: (1) chloride-free buffer (composed of 125 mM sodium gluconate, 4.8 mM potassium gluconate, 1.2 mM magnesium sulfate, 1.3 mM calcium gluconate, and 5 mM HEPES; adjusted to pH 7.4 with sodium hydroxide); or (2) sodium buffer (composed of 140 mM sodium chloride, 4.73 mM potassium chloride, 1.25 mM calcium chloride, 1.25 mM magnesium sulfate, and 5 mM HEPES, adjusted to pH 7.4 with sodium hydroxide); or (3) sodium-free buffer (composed of 140 mM N-methyl-D-glucamine chloride, 1.25 mM magnesium sulfate, 4.73 mM potassium chloride and 1.25 mM calcium chloride), adjusted to pH 7.4 with potassium hydroxide). For trans-stimulation studies, the experimental conditions described in our previously published methods were followed[2,3]. In brief, the SLC22A10 or EV stable cell lines were pre-incubated with either buffer or 2 mM succinic acid, 2 mM α-ketoglutaric acid, 2 mM butyric acid, or 2 mM glutaric acid for 2 hours. Subsequently, the cells were washed twice with HBSS before commencing the uptake of the anions (estradiol-17β-glucuronide). All transporter uptake assays were replicated in at least one additional experiment to confirm the results. All results shown in figures are from a representative experiment with three or four technical replicates.

### Kinetic studies of estradiol glucuronide

Kinetic studies of estradiol-17β-glucuronide were conducted in HEK293 Flp-In cells expressing chimpanzee SLC22A10 isoforms (533 amino acid and 552 amino acid) that were stably transfected. The kinetic studies described below followed the methods published by our research group[2,3,23]. Initially, we examined the time-dependent uptake of the substrates using trace amounts of the radioactive compound. Concentrations of the non-labeled compounds were varied up to 50 μM. For the kinetic studies, a duration of five minutes at 37 °C was chosen as it fell within the linear range observed in the uptake versus time plot for each substrate. Each data point represents the mean ± standard deviation of uptake in the cells transfected with the transporter, subtracted by that in empty vector cells. The obtained data were fitted to a Michaelis-Menten equation to estimate the kinetic parameters. Plots were generated based on a representative experiment out of three independent studies.

### Protein extraction and global proteomics of HEK293 cells expressing SLC22A10 orthologs

HEK293 Flp-In cells were transfected transiently with various SLC22A10 orthologs, including human, chimpanzee, and the mutations to proline or leucine at position 220. After 72 hours of transfection, cell pellets were collected and shipped to Dr. Per Artursson's laboratory in Uppsala University for protein quantification. The quantification was performed on both HEK293 cells and HEK293 cells expressing the different SLC22A10 orthologs and mutations. HEK293 cell pellets (50–92 mg) were lysed in a lysis buffer containing 50 mM dithiothreitol, 2% sodium dodecyl sulfate in 100 mM Tris/HCl pH 7.8. The lysates were incubated at 95 °C for 5 min and sonicated with 20 pulses of 1 second, 20% amplitude by using a sonicator coupled with a microtip probe. The lysates were centrifuged at 14,000×g for 10 min and supernatants were collected. Using LysC and trypsin, the multi-enzyme digestion filter-aided sample preparation (MED-FASP) approach was performed[58]. C18 stage tips were used to desalt the peptide mixture[59,60] and samples were stored at −20 °C until analysis. Protein and peptide content were determined by using tryptophan fluorescence assay[61]. The global proteomics analysis was performed on a Q Exactive HF mass spectrometer (Thermo Fisher Scientific) coupled to a nano–liquid chromatography (nLC). EASY-spray C18-column (50 cm long, 75 μm inner diameter) was used to separate peptides on a ACN/water gradient (with 0.1% formic acid) over 150 min. MS was set to data dependent acquisition with a Top-N method (full MS followed by ddMS2 scans). Proteins were identified using MaxQuant software (version 2.1.0.0)[62] with the human proteome reference from Uni-ProtKB (October, 2022). Total protein approach was used as the protein quantification method[63]. The minimum number of unique peptides for quantification is more than 2 unique and razor peptides. The proteomics data generated in this study have been deposited in the PRIDE database under accession code PXD047102.

### RNA isolation and quantitative RT-PCR

HEK293 Flp-In cells were cultured in poly-D-lysine coated 24-well plates at a seeding density of 1.5-1.8 ×10$^5$ cells per well, allowing them to reach 75-80% confluency. The RT-PCR method for transcript levels determination as detailed bleow, are methods we have previously described[3]. Once the desired confluency was achieved, the cells were transiently transfected with either the vector alone or the vector containing different SLC22A10 orthologs (in the pcDNA3.1(+) expression vector). For the transfection mixture, 500 ng of plasmid DNA, 1 μL of Lipofectamine LTX (Thermo Fisher Scientific), and 100 μL of Opti-MEM I reduced serum media (Thermo Fisher Scientific) were used. After 36-48 hours of transfection, the media was removed, and RNA Lysis buffer (350 μL) was added to each well. Total RNA was isolated from the cells using the Qiagen RNeasy kit (Qiagen). Subsequently, cDNA was synthesized using the SuperScript VILO cDNA Synthesis Kit (ThermoFisher Scientific). For quantitative RT-PCR (qRT-PCR), Taqman reagents and specific primer and probe sets were used, targeting human SLC22A10 (Assay ID: Hs01397962_m1) and beta actin (Assay ID: Hs99999903_m1) (Applied Biosystems, Foster City, CA). The qRT-PCR reactions were conducted in a 96-well plate, with a reaction volume of 10 μL, using the QuantStudio™ 6 Flex Real-Time PCR System and the default instrument settings. The expression levels were determined using the Ct method, and the data were normalized to the endogenous levels of beta actin. The results are presented as fold-increases in the SLC22A10 transcript levels relative to the cell lines expressing the vector control.

### Cloning of SLC22A10 in pooled human liver

For the cloning process, pooled total RNA samples from human liver were obtained from Clontech. Each sample (2 μg) of total RNA was reverse transcribed into cDNA using the SuperScript VILO cDNA Synthesis kit (Thermo Fisher Scientific) following the manufacturer's instructions. The primers specified below were employed for PCR amplification of the NM_001039752 transcript: Forward primer: ACCGAGCTCGGATCCATGGCCTTTGAGGAGCTC; and reverse primer: CCCTCTAGACTCGAGTTATGCCTTTTCCTTGAGATT. The nucleotide underlined are open reading frame of SLC22A10. The resulting PCR products were cloned into BamHI and XhoI multiple-cloning site of the pcDNA5FRT vector and subsequently subjected to sequencing at MCLAB in South San Francisco to determine the sequence of the transcript. For the cloning of human SLC22A10, the KOD Xtreme Hot

Start DNA polymerase kit (Takara) was utilized. The PCR cycling conditions were as follows: (i) initial activation at 94 °C for 2 minutes, (ii) denaturation at 98 °C for 10 seconds, (iii) annealing at 57.5 °C for 30 seconds, and (iv) extension at 68 °C for 1 minute.

## Calculating ΔΔG with the PRISM's rosetta_ddG_pipelne v 0.2.4[21] using Rosetta v 3.15

In brief, the full-length human SLC22A10-P220 (Uniprot ID Q63ZE4) structure predicted by the AlphaFold[64] from the AlphaFold DB[65] was oriented in the membrane with PPM 3.0 web server[66] with the default settings and relaxed using the Rosetta's relax protocol[66]. The best of 20 generated structures was used for the ΔΔG calculations with the cartesian_ddG protocol[67] repeated in 5 replicas. ΔΔE scores were calculated using GEMME[68] and rank-normalized as in the reference paper[21]. The relaxed structure, ΔΔG, and ΔΔE results are available at zenodo, [https://doi.org/10.5281/zenodo.8411757].

## Flow cytometry analysis

To determine if human SLC22A10 reference (Pro220) is rapidly degraded upon biosynthesis, HEK293 Flp-In cells stably transfected with SLC22A10 reference (Pro220) or SLC22A10 mutant (Leu220) in the pcDNA3.1(+)-NGFP, or an empty vector control and seeded in a 24-well cell-culture plate. When the cells had reached approximately 70 % confluence, they were treated with 30 nM of the 26 S proteosome inhibitor bortezomib (Millipore Sigma (CAS 179324-69-7)). After 16 hours the cells were washed and analyzed by flow cytometry on an Attune NxT Acoustic Focusing Cytometer (Life Technologies). Data was analyzed using the Flowjo software package and GFP fluorescence used to determine the extent of protein accumulation upon proteosome inhibition.

## Transporter uptake studies and LC-MS/MS analysis

The list of steroid conjugates and their sources can be found in the Supplementary Table 2. Each steroid conjugates was dissolved in DMSO to obtain 20 mM stock solution. Compounds were stored in −20 °C freezer. HEK293 Flp-In cells stably transfected with GFP only, chimpanzee SLC22A10 (533 amino acid) and chimpanzee SLC22A10 (552 amino acid) were plated in poly-D-lysine coated 48-well plates at a seeding density of 1.5 ×10^5 cells per well, allowing them to reach 90-95% confluency after 16-24 hours. Prior to uptake studies, the culture medium (Dulbecco's modified Eagle's medium, DMEM) supplemented with 10% fetal bovine serum was aspirated, and the cells were incubated in 0.8 mL of Hank's Balanced Salt Solution (HBSS) at 37 °C for 10-20 minutes. To screen various steroid and steroid conjugate compounds as SLC22A10 substrates, HEK293 Flp-In cells stably expressing GFP or chimpanzee SLC22A10 were incubated with HBSS buffer containing 10 μM of the respective compounds for 20 minutes. The uptake reactions were terminated by washing the cells twice with 0.8 mL of HBSS buffer, followed by incubation in 400 μL of methanol. After 30 minutes of shaking at room temperature, 300 μL of methanol containing the extracted steroid or steroid conjugates from each well were transferred to a 1.5 mL tube and stored at −80 °C before quantification using LC-MS/MS analysis.

Experimentally, cells in culture were presented with commercially available steroid metabolites and their presence in isolated cells was quantified against authentic standards. Subaliquot of methanol disrupted cellular extracts (90 μL) were spiked with 10 μL of deuterated 17β-estradiol glucuronide, mixed by vortexing, and filtered at 0.2 μm through polyvinyl difluoride membranes (Agilent Technologies, Santa Rosa, CA, USA) by centrifugation and 10,000 g. After filtration, samples were enriched with 25 nM 1-cyclohexyl-3-uriedo-decanoic acid (Sigma-Adlrich, St. Lousi MO) as an internal standard. Metabolites were measured using ultra-performance liquid chromatography–electrospray ionization tandem mass spectrometry (UPLC-ESI-MS/MS) on a API 4500 QQQ (Sciex, Framingham, MA) with a scheduled multiple reaction monitoring (MRM) using methods adapted from Ke et.al, 2015[69]. Analytes were separated on a Waters I-Class UPLC-FTN equipped with a 2.1 × 100 mm i.d., 1.7 μm Acquity BEH C$_{18}$ column (Waters Co; Milford, MA) held at 50 °C. Analytes in 5 μL injections were separated using water (solvent A) and methanol (solvent B) both containing 2 mM ammonium formate at 400 μL/min with the following gradient: Initial 40% B to 70% B at 2 min, to 98%B at 3 min, held to 4 min, to 40%B at 5.1 min held to 6 min. Data were processed using MultiQuant v. 3.02 (Sciex), with five-point calibration curves bracketing the observed concentration ranges fit with quadradic curves using 1/x weighting for each steroid conjugate (Supplementary Fig. 10). Mass spectrometer acquisition parameters and analyte retention times are described in Supplementary Table 3.

## Sequencing data processing and analyses of SLC22A10 in greater apes

Orthologous genome regions of *SLC22A10* coding sequence in multiple primate species were obtained using UCSC liftOver (default parameters)[70] based on hg38 (human), panTro6 (chimpanzee), panPan3 (bonobo), gorGor6 (gorilla) and ponAbe3 (orangutan) assemblies. These regions were aligned using MUSCLE[71] and visualized using MView[72]. *SLC22A10* exons and coding sequences are based on RefSeq annotations[73] for human, chimpanzee and gorilla. Despite the long *SLC22A10* isoform not being annotated in RefSeq for bonobo and orangutan, we included these two species in the alignment considering the plausibility of the protein model in the context of their genomes and the high genomic conservation in comparison to chimpanzee and gorilla.

Allele frequencies from non-human great apes were obtained from available whole-genome sequencing data including 59 chimpanzees, 10 bonobos, 49 gorillas and 16 orangutans[74–77]. All samples were mapped to Hg19. We extracted the genotyping information in position chr11:63078478 (Hg19 coordinates) to calculate the allele frequencies per population. We manually curated the genotypes by checking the raw reads overlapping this region in the BAM files. In chimpanzees we report a frequency of the insertion to be 3/108 = 2.5%; in bonobos is 0/20 = 0%; in gorillas is 0/98 = 0%; and in orangutans is 0/32 = 0%. The global human allele frequencies were obtained from 1000 Genomes Project Phase 3[78] database in Ensembl for the rs562147200 SNP.

## SLC22A10 variants in archaic hominins and ancient humans

Archaic hominin genotypes for three Neanderthals and a Denisovan were retrieved from [http://ftp.eva.mpg.de/neandertal/Vindija/VCF/] [and [http://ftp.eva.mpg.de/neandertal/Chagyrskaya/VCF/][15–18]. We used the LDproxy tool from LDlink[79] to identify variants in high LD with rs1790218 in all Thousand Genomes populations and intersected these variants with those assayed in ancient humans from the Allen Ancient DNA Resource (AADR)[19], retrieved from [https://reichdata.hms.harvard.edu/pub/datasets/amh_repo/curated_releases/V54/V54.1.p1/SHARE/public.dir/v54.1.p1_1240K_public.tar]. We identified two such variants: rs1783634 (D' = 1, r2 = 0.9839) and rs1201559 (D' = 0.9975, r2 = 0.9775). In the AADR analyses, we excluded samples from archaic hominins and individuals that were not genotyped at both loci. For modern samples present in duplicate in the AADR, we filtered out genotypes called from pseudo-haploid sequences and retained a single genotype per SNP per individual from samples with coverage sufficient for calling diploid genotypes (N = 2,610). Among the remaining duplicated samples, we retained genotypes for each individual where the genotypes from the duplicated samples matched at both proxy SNPs (N = 83). Genotypes that did not match between duplicated samples were removed from consideration (N = 11). This resulted in genotypes from 5,045 individuals (1,375 ancient and 3,670 modern). We calculated allele frequency in 17 time periods, stratifying by five continental groups defined based on sampling locality: Africa, Americas, Europe,

Asia, and Oceania. We also retrieved the allele age estimate for rs1790218 using the Human Genome Dating tool[80] with all default settings. We used the LDhap tool LDlink to quantify haplotypes for rs1790218 and rs562147200 in all Thousand Genomes populations. The archived version of the code used to analyze ancient human genomes and visualize allele frequencies has been deposited in Zenodo [https://doi.org/10.5281/zenodo.10823210]. A non-archived version is available on GitHub [https://github.com/brandcm/SLC22A10_Variant_Evolutionary_History].

## Statistical analysis

When comparing the significant differences among HEK293 cells transfected with GFP only and various SLC22A10 ortholog species or other transporters, we performed multiple comparisons using one-way analysis of variance followed by Dunnett's two-tailed test. HEK293 cells transiently transfected with the GFP vector served as the control. The fold uptake of the substrate, relative to the control cells, was plotted based on one representative experiment conducted in triplicate wells (mean ± s.d.). Statistical significance was indicated as ***$p < 0.0001$, **$p < 0.01$, *$p < 0.05$. These findings were further confirmed through at least one or two additional experiments. For specific differences and more detailed information, please refer to the figure legend.

## Reporting summary

Further information on research design is available in the Nature Portfolio Reporting Summary linked to this article.

## Data availability

The global proteomics data for HEK293 Flp-In cell lines, stably transfected with human and chimpanzee SLC22A10 in both reference and mutant forms, are available at PRIDE (Proteomics IDEntifications Database), https://www.ebi.ac.uk/pride/. The project's accession number is PXD047102. Archaic hominin genotypes for three Neanderthals and a Denisovan were retrieved from [http://ftp.eva.mpg.de/neandertal/Vindija/VCF/] and [http://ftp.eva.mpg.de/neandertal/Chagyrskaya/VCF/]. Ancient human genotypes from the Allen Ancient DNA Resource (AADR) were retrieved from this site [https://reichdata.hms.harvard.edu/pub/datasets/amh_repo/curated_releases/V54/V54.1.p1/SHARE/public.dir/]. The sequencing data of SLC22A10 in greater apes are obtained from these accession code: PRJEB15086 [https://www.ebi.ac.uk/ena/browser/view/PRJEB15086], PRJEB19688, PRJNA189439. The silhouette images of primates in Fig. 2c were obtained from PhyloPic [https://www.phylopic.org/], which are available for reuse under Creative Commons [https://creativecommons.org/] licenses. Source data are provided with this paper.

## Code availability

The archived version of the code used to determine mutation effect predictions for the P220L variant of human SLC22A10 transporter has been deposited in Zenodo [https://doi.org/10.5281/zenodo.8411757]. The archived version of the code used to analyze ancient human genomes and visualize allele frequencies has been deposited in Zenodo [https://doi.org/10.5281/zenodo.10823210]. A non-archived version is available on GitHub [https://github.com/brandcm/SLC22A10_Variant_Evolutionary_History].

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

## Acknowledgements

This research was supported by NIH GM117163 (SWY) and GM139875 (KMG, SWY) and EY032161 (JEG). Additional support was provided by the Swedish Research Council (0151) (M.C, N.H, P.A) USDA Intramural Projects 2032-51530-025-00D (JWN) and by Mary Anne Koda-Kimble Seed Award for Innovation (2020, 2021) (SWY). The USDA is an equal opportunity employer and provider. TMB is supported by funding from the European Research Council (ERC) under the European Union's Horizon 2020 research and innovation program (grant agreement No. 864203), PID2021-126004NB-100 (MICIIN/FEDER, UE) and Secretaria d'Universitats i Recerca and CERCA Programme del Departament d'Economia i Coneixement de la Generalitat de Catalunya (GRC 2021 SGR 00177). We would like to acknowledge Esther Lizano, Alba Duch and Mina Azimi for their invaluable assistance and support in the successful execution of our research.

## Author contributions

S.W.Y and K.M.G. conceived and designed the research. S.W.Y, K.M.G. wrote the main text of the manuscript. S.W.Y., G.L., X.Z., J.Y. performed transporter and/or kinetic studies. S.W.Y. generate cell lines expressing SLC22A10 orthologs, performed RNA isolation and performed data analysis for the transporter and kinetic studies. H-C.C. performed the cloning of SLC22A10. M.L.K. performed the fluorescence microscopy and imaging analysis. C.F. and L.F-P. performed sequencing data processing and analyses of SLC22A10 in greater apes. T.M-B. helped interpret and discuss the results of the sequencing analysis of the SLC22A10 in greater apes. M.C. performed protein isolation, analyzed the proteomic data. N.H. helped interpret and analysis of the proteomic data. M.C. and N.H. wrote the main text of the manuscript related to the proteomic experiment. V.M.A. performed the LC-MS/MS analysis. J.W.N. helped interpret the analysis of the LC-MS/MS. V.M.A and J.W.N. wrote the main text of the manuscript related to the LC-MS/MS analysis. C.M.B. and J.A.C provided the data on SLC22A10 variants in archaic hominins and ancient humans, interpreted and discussed the results and wrote the main text of the manuscript related to the SLC22A10 variants in archaic hominins and ancient humans. A.Z. provided data on stability calculations of human SLC22A10 and mutants. A.Z. and A.S. wrote the main text of the manuscript related to the stability calculations of human SLC22A10 and mutants. F.D.W. performed the flow cytometry study and analyzed the result. F.D.W and J.E.G wrote the main text of the manuscript related to the flow cytometry analysis. S.W.Y., L.F-P., C.F., M.C., M.L.K., N.H., V.M.A., A.Z., C.M.B., F.D.W., J.A.C., J.W.N. analyzed the results. S.W.Y., L.F-P., P.A-M., C.F., M.C., M.L.K., N.H., V.M.A., J.D., A.Z., A.S., C.M.B., F.D.W., J.E.G., J.A.C., P.A., J.W.N., T.M-B and K.M.G. interpreted and discussed the results. All authors revised the manuscript.

## Competing interests

The authors declare no competing interests.

## Additional information

[1]Department of Bioengineering and Therapeutic Sciences, University of California, San Francisco, CA, USA. [2]Institute of Evolutionary Biology (UPF-CSIC), PRBB, Dr. Aiguader 88, 08003 Barcelona, Spain. [3]Center for Evolutionary Hologenomics, The Globe Institute, University of Copenhagen, Øster Farimagsgade 5A, 1352 Copenhagen, Denmark. [4]Department of Pharmacy, Uppsala University, Uppsala, Sweden. [5]United States Department of Agriculture, Agricultural Research Service, Western Human Nutrition Research Center, Davis, CA 95616, USA. [6]Joint Research Unit for Infectious Diseases and Vectors Ecology Genetics Evolution and Control (MIVEGEC), University of Montpellier, French National Center for Scientific Research (CNRS 5290), French National Research Institute for Sustainable Development (IRD 224), 911 Avenue Agropolis, BP 64501, 34394 Montpellier Cedex 5, France. [7]Department of Pharmaceutical Chemistry, University of California, San Francisco, CA, USA. [8]Quantitative Biosciences Institute (QBI), University of California, San Francisco, San Francisco, CA, US. [9]Bakar Computational Health Sciences Institute, University of California, San Francisco, CA, USA. [10]Department of Epidemiology and Biostatistics, University of California, San Francisco, CA, USA. [11]Department of Ophthalmology, University of California, San Francisco, CA, USA. [12]Institute for Neurodegenerative Diseases, University of California, San Francisco, CA, USA. [13]Institute for Human Genetics, University of California San Francisco, San Francisco, CA, USA. [14]Science for Life Laboratories, Uppsala University, Uppsala, Sweden. [15]Department of Nutrition, University of California, Davis, Davis, CA 95616, USA. [16]Catalan Institution of Research and Advanced Studies (ICREA), Passeig de Lluís Companys, 23, 08010 Barcelona, Spain. [17]CNAG, Centro Nacional de Analisis Genomico, Barcelona Institute of Science and Technology (BIST), Baldiri i Reixac 4, 08028 Barcelona, Spain. [18]Institut Català de Paleontologia Miquel Crusafont, Universitat Autònoma de Barcelona, Edifici ICTA-ICP, c/ Columnes s/n, 08193Cerdanyola del Vallès, Barcelona, Spain. ✉e-mail: kathy.giacomini@ucsf.edu

