## [Peer Review File · Nature Communications]

Illuminating the Function of the Orphan Transporter,
SLC22A10, in Humans and Other PrimatesReviewers' Comments:

Reviewer #1:

Remarks to the Author:

There are many poorly understood SLC transporters. *slc22a10* is part of a group of transporters that, together, seem highly associated with sex steroids by association studies, sequence analysis and clustering. The story of at least one member (*slc22a24*) of the group appears well worked out. Here it is reported that the *slc22a10* transporter is inactivated in humans due to a missense mutation, but is active in primates. The work seems well done.

Major Concerns

It is hard to understand the broader significance to human biology. The direct connection to key steps between this transporter the well developed field of steroid hormone metabolism is not clearly described. Has the connection been established?

The extended gene cluster has a number of *slc22* transporters, which seems quite large in certain species like rodents. It seems that many of these have a strong selectivity for conjugated sex steroids. Though some distance away, *slc22a8*, while being a high affinity steroid transporter, appears in the literature to be a transporter of drugs, toxins and other various metabolites. Nevertheless, is it clear that it is unimportant for sex steroid metabolism?

According to the Discussion, *slc22a20* seems to be a similar gene no longer active in humans. It is also in the same family (*slc22*), and one of its few known transported substrates in assays is a conjugated sex steroid (PMID: 20519554). It seems suited to transporting sex steroids across the blood-testes barrier.

It seems important to clarify how loss of functional transporters impacts human steroid biology or the evolution of some key steroid-dependent mechanism? For *slc22a10*, the authors gesture to the liver, but there must be more.

Reviewer #2:

Remarks to the Author:

Solute carrier (SLC) transporters are a large group of membrane proteins with a significant potential role in human physiology. About 30% of SLC transporter are currently "orphans", that is they have no known substrate. For a better understanding of their physiological role, it is therefore important to identify their function.

In the present study, the authors have deorphaned human SLC22A10. They combined various methods and techniques, e.g. sequence comparisons with other primates, in vitro transport assays, immunolocalization of GFP-tagged proteins, site-directed mutagenesis analyses and various in silico analyses tools. In general, the authors could show that human SLC22A10 is a pseudogene and not functional in humans, but functions as a steroid glucuronide transporter in great apes. Human SLC22A10 contains a disrupting mutation at position 220, which leads to Pro220 - instead of Leu220 in the great apes - and a non-functional protein not located at the plasma membrane. Site-directed mutagenesis of Pro220 to Leu220 resulted in restoration of a functional human SLC22A10 located in the plasma membrane.

This manuscript describes an interesting novel strategy to deorphan SLC transporters in general. The results are important for a better understanding of SLC22A10 function and for understanding evolution of this gene.

I have a few comments:

- Humans have an A nucleotide insertion at the first base pair of exon 10 (rs562147200) with an allele frequency of about 98% in all human populations in the gnomAD database. This insertion should lead to a 541 amino acid protein. According to the authors' analysis, this A nucleotide insertion is apparently sufficient to retain SLC22A10 intracellularly (Fig. 1B). The authors also convincingly show that variant Pro220, which was gained in the hominid evolution, is a disrupting mutation. Is it possible to tell whether the A nucleotide insertion (rs562147200) arose before or after occurrence of Pro220? Would the A nucleotide insertion in this respect be similar to the prevalent nonsense variant Trp96Ter?

Some minor points:

- Results of Fig.4 A and B should be described in the Results section.

- Legend to supplementary Fig. 4: (i) "sqm" should be defined; (ii) what is meant by "All expression vectors do not have GFPtagged."

- line 290: check "chimpanzee SLC22A22A10"

- line 294: define in which section of <https://www.proteinatlas.org> it can be found that SLC22A10 is not expressed in human liver tissue

- line 399: it should be SLC22A24 instead of SLC2224

- line 493: two vectors are given for tagging of SLC22A10 with GFP. Which one was used? pcDNA3.1(+)-eGFP, or pcDNA3.1(+)-N-eGFP?

- line 621: provide reference

- line 690: the link to zenodo should be given

- line 720, Table S2: check spelling of 1-cycloheyl-3-uriedo-decanoic acid

Reviewer #3:

Remarks to the Author:

In this interesting work, Giacomini and colleagues present evidence that the orphan transporter SLC22A10 was rendered nonfunctional during human evolution, whereas corresponding orthologs in great apes transport estradiol-17 β -glucuronide. Remarkably, this difference in function is caused by just one mutation on position 220 (L to P). All the experiments conducted - including mutating Pro220 in human SLC22A10 to Leu and mutating position 220 in Chimpanzee SLC22A10 to Pro - convincingly support the presented hypothesis.

Analysing an AlphaFold2 model of human SLC22A10 further indicates that the kink produced by Pro exposes a lysine, which might induce ubiquitination. However, there is no further experimental evidence for this presented. I thus wonder if the authors could obtain further experimental data that indeed enhanced degradation of SLC22A10 takes place.

REVIEWER COMMENTS

Reviewer #1 (Remarks to the Author):

There are many poorly understood SLC transporters. slc22a10 is part of a group of transporters that, together, seem highly associated with sex steroids by association studies, sequence analysis and clustering. The story of at least one member (slc22a24) of the group appears well worked out. Here it is reported that the slc22a10 transporter is inactivated in humans due to a missense mutation, but is active in primates. The work seems well done.

Response: Thank you for taking the time to provide thoughtful review of our work on the SLC22A10 transporter.

Major Concerns

It is hard to understand the broader significance to human biology. The direct connection to key steps between this transporter the well developed field of steroid hormone metabolism is not clearly described. Has the connection been established?

Response: Thank you for your insightful question. The connection between the membrane transporter field and the field of steroid hormone metabolism is very important. In response to the reviewers comments, we have included a new paragraph in our discussion, which describes these relationships and emphasizes the broader relevance to human biology. In addition, we also included statement in the last paragraph in the Discussion to state that future studies are needed to understand the role of SLC22A10 in great apes.

Line 417-431 (Please note that the references for the response to the reviewer's comment can be found at the end of the response page.)

“Our fifth finding suggests that SLC22A10 plays a significant role in the liver and provides some insight into the biological role of the transporter. RNA-seq data indicate that the SLC22A10 gene is specifically expressed in the liver in chimpanzees, bonobos, and gorillas, with very low expression in the brain¹, mirroring the transcriptomic data in the human GTEx project. The liver is a major organ for steroid hormone metabolism, such as UDP-glucuronosyltransferases (UGT), which forms steroid glucuronide conjugates to enhance water-solubility for excretion by the kidney⁵. Similarly, the liver is also abundant in beta-glucuronidase⁶, which removes glucuronides and forms the parent compound, such as steroids. Notably, several steroid conjugate influx and efflux transporters are expressed in the liver, some of which have greater specificity for steroid conjugates and many other substrates (e.g., SLCO1B1, SLCO1B3), while others have narrow specificities (e.g., SLC22A9, known to transport sulfate conjugates^{3, 7}). SLC22A10 in great apes exhibited significant uptake of steroid glucuronides (Fig. 5, Fig. S1) but not other well-known anions, which are substrates of other SLC22 family members, such as bile acids and dicarboxylic acids (Fig. 1, Fig. S1, Fig. S8). To our knowledge, SLC22A10 is the only specific steroid glucuronide transporter in the liver, possibly due to its specialized role in the uptake of steroid conjugates in the liver to expose them to beta-glucuronidase, forming the parent steroid.”

Line 437-439 and Line 465-469

“Overall, the broader impact of this study on SLC22A10, an orphan transporter, is multifaceted and significant. The study not only sheds light on the specific characteristics of SLC22A10 but also contributes to a broader understanding of genetic evolution, the role of orphan transporters in biological processes, and potential applications in medicine and pharmacology. Future studies are needed to determine whether the loss of function human SLC22A10-P220 is a favorable situation for humans and whether similar mechanisms have led to the inactivation of other orphan genes in the human genome. In addition, studies are needed to determine the role of SLC22A10 in biological processes such as the transport of sex steroid conjugates in great apes.”

The extended gene cluster has a number of slc22 transporters, which seems quite large in certain species like rodents. It seems that many of these have a strong selectivity for conjugated sex steroids. Though

some distance away, slc22a8, while being a high affinity steroid transporter, appears in the literature to be a transporter of drugs, toxins and other various metabolites. Nevertheless, is it clear that it is unimportant for sex steroid metabolism?

Response: The reviewer is correct in highlighting that many of the transporters in the cluster have a pronounced selectivity for conjugated sex steroids. Regarding SLC22A8, its primary role is not typically associated with sex steroid metabolism due to its capacity to transport a diverse set of molecules with different affinities^{2, 3}. Research involving knockout mice, specifically comparing plasma levels between wild-type and knockout subjects, indicates a spectrum of non sex steroid or steroid conjugates that are significantly elevated in the plasma of knockout mice or reduced in urinary concentrations (e.g. 2-oxoglutarate, pongamoside A, dehydroascorbic acid)^{8, 9}. Additionally, kynurenic acid is a key biomarker for SLC22A8¹⁰. We've adjusted the text to delineate the differences between SLC22A8 and the other genes in the cluster. Line 353-357 (Please note that the references for the response to the reviewer's comment can be found at the end of the response page.)

"In contrast, SLC22A8, a multispecific transporter, expressed primarily in the kidney, for an array of drugs, toxins, and other metabolites, is not typically associated with sex steroid metabolism due to its capacity to transport a diverse set of molecules with varying affinities^{9, 11}. In Slc22a8 null mice, a spectrum of non-sex steroid conjugates was shown to be significantly elevated in the plasma⁸⁻¹⁰.

According to the Discussion, slc22a20 seems to be a similar gene no longer active in humans. It is also in the same family (slc22), and one of its few known transported substrates in assays is a conjugated sex steroid (PMID: 20519554). It seems suited to transporting sex steroids across the blood-testes barrier.

Response: Thank you for highlighting the intriguing paper on the role of SLC22A20, a sex steroid conjugate transporter expressed at the blood-testes barrier. This reference provides valuable context, and we include this reference in the discussion along with the following sentence (Line 463 – 465):

"There are known pseudogenes in the SLC superfamily, for example, SLC22A20, functions as a steroid conjugate transporter (esterone sulfate) in mice and is expressed in the blood-testes barrier¹²."

It seems important to clarify how loss of functional transporters impacts human steroid biology or the evolution of some key steroid-dependent mechanism? For slc22a10, the authors gesture to the liver, but there must be more.

Response: We appreciate the reviewer's observation. To address the impact of the loss of functional transporters on human steroid biology and the evolution of key steroid-dependent mechanisms, we investigated a few datasets. In examining a data set of tissue samples of individual non-human primate species available at NHPRTR (<http://www.nhprtr.org/>, http://www.nhprtr.org/data/2014_NHP_tissuespecific/), it seems that the expression of SLC22A10 mRNA is specific to the liver in great apes. This expected profile aligns closely with the expression data for humans as documented in GTEx, <https://www.gtexportal.org/home/gene/SLC22A10>. In addition, we were able to obtain RNAseq data from another publication¹³, which has RNAseq analysis using tissue samples from human, chimpanzee, bonobo and gorilla. The above-mentioned data clearly show that SLC22A10 is specifically expressed in the liver (see Figure 1 for Reviewer #1). Since steroid metabolism to form glucuronides mainly occurs in the liver, these data support the idea that SLC22A10 is involved primarily in hepatic steroid metabolism. The evolutionary reason for which human SLC22A10 was lost is unknown but is likely related to divergent evolutionary pathways in steroid metabolism between humans and great apes, which needs further study.

Similarly, in humans, the data suggest that SLC22A10 is primarily expressed in the liver, though there is some evidence that it may also be expressed in neurons.

- (i) RNA Expression, <https://www.proteinatlas.org/ENSG00000184999-SLC22A10/tissue>: Several RNAseq databases indicate SLC22A10 is specifically expressed in the liver. The result from GTEx are shown in Figure 1 below.
- (ii) RNA Expression in different brain region, <https://www.proteinatlas.org/ENSG00000184999-SLC22A10/brain>: Human Protein Atlas Human brain dataset showed expressions of SLC22A10 in different brain region, TPM values 1 to 7. However, GTEx and FANTOMS human brain dataset did not show significant expressions of SLC22A10 transcript.

Single cell, <https://www.proteinatlas.org/ENSG00000184999-SLC22A10/single+cell+type>: In this dataset, the data also showed enrichment of SLC22A10 in liver and specifically in the hepatocytes. In addition, the transporter seems to express in the excitatory neurons, and there is one unvalidated association study with neuronal injury in a small cohort (<https://www.ncbi.nlm.nih.gov/snp/rs562147200>).

We added the following sentences in the Discussion (Line 433-435):

“Although the transporter is primarily expressed in liver, it appears to be expressed at lower levels in neurons (https://www.proteinatlas.org/ENSG00000184999-SLC22A10/single+cell+type).”

Reviewer #2 (Remarks to the Author):

Solute carrier (SLC) transporters are a large group of membrane proteins with a significant potential role in human physiology. About 30% of SLC transporter are currently “orphans”, that is they have no known substrate. For a better understanding of their physiological role, it is therefore important to identify their function.

In the present study, the authors have deorphaned human SLC22A10. They combined various methods and techniques, e.g. sequence comparisons with other primates, in vitro transport assays, immunolocalization of GFP-tagged proteins, site-directed mutagenesis analyses and various in silico analyses tools. In general, the authors could show that human SLC22A10 is a pseudogene and not functional in humans, but functions as a steroid glucuronide transporter in great apes. Human SLC22A10 contains a disrupting mutation at position 220, which leads to Pro220 - instead of Leu220 in the great apes - and a non-functional protein not located at the plasma membrane. Site-directed mutagenesis of Pro220 to Leu220 resulted in restoration of a functional human SLC22A10 located in the plasma membrane.

This manuscript describes an interesting novel strategy to deorphan SLC transporters in general. The results are important for a better understanding of SLC22A10 function and for understanding evolution of

this gene.

Response: We appreciate your above comments.

I have a few comments:

- Humans have an A nucleotide insertion at the first base pair of exon 10 (rs562147200) with an allele frequency of about 98% in all human populations in the gnomAD database. This insertion should lead to a 541 amino acid protein. According to the authors' analysis, this A nucleotide insertion is apparently sufficient to retain SLC22A10 intracellularly (Fig. 1B). The authors also convincingly show that variant Pro220, which was gained in the hominid evolution, is a disrupting mutation. Is it possible to tell whether the A nucleotide insertion (rs562147200) arose before or after occurrence of Pro220?

Response: This is an interesting question. It is difficult to ascertain whether the A nucleotide insertion arose before or after Pro220 as we could not find sequence information in archaic and ancient humans to provide support for the insertion arising in humans. However, the high frequency of the major allele (an A insertion) and the geographic distribution of the minor allele (largely to a subset of individuals in sub-Saharan Africa) suggest an overall ancient origin. Another line of supporting evidence for old allele age is the strong correlation between alleles at this locus and alleles at loci up to 34,372 bp away, calculated using the LDproxy tool in LDlink. We have included a qualitative age estimate for the insertion in the main text (see lines 212-224). We speculate that given the proline is fixed and its functional effect, we believe it came first but it is difficult to prove this.

Would the A nucleotide insertion in this respect be similar to the prevalent nonsense variant Trp96Ter?

Response: We found using LDproxy in LDlink that the A insertion is not in linkage disequilibrium with the 96Ter, that is they are not on the same haplotype. We now include information on the LD pattern to the supplementary figures (**Table S1**) and we provide some discussion in the Result section. See lines 224-231.

Some minor points:

- Results of Fig.4 A and B should be described in the Results section.

Response: We have inserted the following sentences in the Result section "Analyses of SLC22A10 isoforms in chimpanzee, bonobo, orangutan and gibbon" (Line 279 – 295).

"The shorter isoforms are derived from alternative acceptor sites (chimpanzee and bonobo) and exon extensions (orangutan and gibbon) resulting in 533-amino acid proteins (see **Fig. 4A**, Supplementary Fig. 7). The HEK293 Flp-In cells that were stably transfected with GFP-conjugated chimpanzee SLC22A10, consisting of 533 amino acids, showed a clear plasma membrane localization. However, it appears that the bonobo SLC22A10, which consists of 538 amino acids, exhibited weaker localization, and the orangutan SLC22A10, also comprising 533 amino acids, showed mixed localization (**Fig. 4B**). However, the uptake by bonobo SLC22A10, comprising 538 amino acids, is not significant, and there is a weaker fold uptake by orangutan SLC22A10, consisting of 533 amino acids, consistent with its mixed localization on the plasma membrane. (**Fig. 4B** and **4C**)."

- Legend to supplementary Fig. 4: (i) "sqm" should be defined; (ii) what is meant by "All expression vectors do not have GFPtagged."

Response: Because of the additional data included in this revised manuscript. The previous supplementary Fig. 4 is now supplementary Figure 8. Below we pasted the additional information inserted in Supplementary Figure 8.

- (i) SqmSLC22A10-552aa: This is the cDNA sequence of SLC22A10 from the Squirrel Monkey, encoding 552 amino acids (ENSSBOT00000021267.1, Uniprot: A0A2K6SBP5).
- (ii) We meant to say that the expression vectors in the experiment do not have GFP tags. We corrected the sentence to: "The expression vectors used in this experiment are not GFP-tagged". We included the experiments to show that the GFP-tagged SLC22A10 does not affect the uptake of the substrates and consistent with our interpretation.

- line 290: check "chimpanzee SLC22A22A10"

Response: Thank you for pointing this out and we have corrected this.

- line 294: define in which section of <https://www.proteinatlas.org> it can be found that SLC22A10 is not expressed in human liver tissue

Response: Thank you for checking. The link to see the SLC22A10 protein is <https://www.proteinatlas.org/ENSG00000184999-SLC22A10/tissue> and we included this link in the text

- line 399: it should be SLC22A24 instead of SLC2224

Response: Thank you for pointing this out and we have corrected this.

- line 493: two vectors are given for tagging of SLC22A10 with GFP. Which one was used? pcDNA3.1(+)-eGFP, or pcDNA3.1(+)-N-eGFP?

Response: We inserted the cDNA into the multiple cloning sites on the expression vectors pcDNA3.1(+) or pcDNA3.1(+)-N-eGFP. We did not use the pcDNA3.1(+)-C-eGFP.

- line 621: provide reference

Response: We have inserted three references from our research groups which have experimental methods for performing kinetic studies. These are PMID: 29720497, 31553721, 33124720.

- line 690: the link to zenodo should be given

Response: We have included this link in the manuscript, <https://zenodo.org/records/8411757>.

- line 720, Table S2: check spelling of 1-cycloheyl-3-uriedo-decanoic acid

Response: We corrected the spelling to 1-cyclohexyl-3-uriedo-decanoic acid.

Reviewer #3 (Remarks to the Author):

In this interesting work, Giacomini and colleagues present evidence that the orphan transporter SLC22A10 was rendered nonfunctional during human evolution, whereas corresponding orthologs in great apes transport estradiol-17 β -glucuronide. Remarkably, this difference in function is caused by just one mutation on position 220 (L to P). All the experiments conducted - including mutating Pro220 in human SLC22A10 to Leu and mutating position 220 in Chimpanzee SLC22A10 to Pro - convincingly support the presented hypothesis.

Analysing an AlphaFold2 model of human SLC22A10 further indicates that the kink produced by Pro exposes a lysine, which might induce ubiquitination. However, there is no further experimental evidence for this presented. I thus wonder if the authors could obtain further experimental data that indeed enhanced degradation of SLC22A10 takes place.

Response: We greatly appreciate the reviewer's evaluation. To address this question, we invited two additional co-authors, Drs. Finn D. Wolfreys and Jason E. Gestwicki, to discuss and provide guidance on the experiment. Their expertise lies in the use of proteasome inhibitors to target cellular mechanisms of protein degradation by blocking the proteasome.

We took two approaches to address this comment:

- (1) We mentioned in the discussion that "Using AlphaFold2 model of human SLC22A10, we observed that the proline 220 in humans introduces a kink in the alpha-helix, which in turn can affect the conformation of the 225-230 loop and. In addition, based on the $\Delta\Delta G$ calculation the proline 220 is significantly less stable with a less negative value of $\Delta\Delta G$ " To show whether this is the possible mechanism, we performed site directed mutagenesis to mutate lysine at 230 positions to arginine. We determined the uptake of estradiol-17 β -glucuronide and the protein abundance using flow cytometry. As shown in the figure below, the experiment did not show that human SLC22A10 proline 220 with arginine 230 significantly improved transport activity nor increased abundance of the GFP-tagged protein. However human SLC22A10 leucine 220 and leucine 220 with arginine 230 showed increased uptake of estradiol-17 β -glucuronide and increase abundance of the GFP-tagged protein. This experimental data does not fully support the prediction that protein 220 affect the accessibility of the lysine 230 for ubiquitination. Therefore, we proceeded with the second approach below.

Figure to Reviewer #3. (a) Uptake of estradiol-17 β -glucuronide in HEK293 Flp-In cells transiently transfected with pcDNA3.1(+)-N-GFP tagged vector containing human SLC22A10 ORF and the mutants. (b) Histograms of GFP fluorescence in HEK293 Flp-In cells stably transfected with empty vector control, and the four different human SLC22A10 reference and mutants. Overall, the approach of changing lysine 230 (reference) to arginine 230 to prevent the accessibility of lysine 230 for ubiquitination on the SLC22A10 background (P220) did not significantly increase the uptake of the estradiol-17 β -glucuronide substrate or increase GFP abundance.

(2) We tested the effect of a 26S proteasome inhibitor, bortezomib at different concentrations in an initial round of experiments using human SLC22A10 reference (Pro220), and measured the abundance of GFP-tagged protein by flow cytometry. Bortezomib-treated samples showed a significant shift in GFP fluorescence intensity, indicating accumulation of SLC22A10. We then compared HEK293 Flp-In cells stably transfected with SLC22A10 (Pro220), SLC22A10 mutant (Leu220), or an empty vector. The histograms below, derived from flow cytometry analysis, are representative of these experiments and are also presented in Supplemental Figure 6 of this revised manuscript. The histograms of GFP fluorescence in HEK293 Flp-In cells stably transfected with the empty vector control, SLC22A10 L220 mutant, and SLC22A10 reference (P220) GFP fusions were treated with either the DMSO vehicle or 30 nM bortezomib. The increased GFP fluorescence observed upon treatment of SLC22A10 reference (P220) indicates the accumulation of SLC22A10, which would otherwise have been rapidly degraded most likely due to its inherent instability. The blue histogram represents the distribution after treatment with the DMSO vehicle, and the red histogram represents the distribution after treatment with 30 nM bortezomib (see the figure below). We added the above new data in the Results section (see line 252-272) and edited the Discussion section (see line 388-392).

Supplemental Figure 6. Histograms of GFP fluorescence in HEK293 Flp-In cells stably transfected with empty vector control, SLC22A10 L220 mutant, and SLC22A10 reference (P220) GFP fusions treated with

DMSO vehicle and 30 nM bortezomib. Increased GFP fluorescence upon treatment of SLC22A10-P220 indicates accumulation of the SLC22A10 that would otherwise have been rapidly degraded likely a consequence of its inherent instability. The blue histogram represents the distribution after treatment with the DMSO vehicle, and the red histogram represents the distribution after treatment with 30 nM bortezomib for 16 hours.

References cited in this response to reviewers' document

1. Lindberg FA, Nordenankar K, Forsberg EC, Fredriksson R. SLC38A10 Deficiency in Mice Affects Plasma Levels of Threonine and Histidine in Males but Not in Females: A Preliminary Characterization Study of SLC38A10(-/-) Mice. *Genes (Basel)*. 2023;14(4). Epub 2023/04/28. doi: 10.3390/genes14040835. PubMed PMID: 37107593; PMCID: PMC10138244.
2. Laatikainen T. Quantitative studies on the excretion of glucuronide and mono- and disulphate conjugates of neutral steroids in human bile. *Ann Clin Res*. 1970;2(4):338-49. PubMed PMID: 5493463.
3. Shin HJ, Anzai N, Enomoto A, He X, Kim DK, Endou H, Kanai Y. Novel liver-specific organic anion transporter OAT7 that operates the exchange of sulfate conjugates for short chain fatty acid butyrate. *Hepatology*. 2007;45(4):1046-55. Epub 2007/03/30. doi: 10.1002/hep.21596. PubMed PMID: 17393504.
4. Bi YA, Costales C, Mathialagan S, West M, Eatemadpour S, Lazzaro S, Tylaska L, Scialis R, Zhang H, Umland J, Kimoto E, Tess DA, Feng B, Tremaine LM, Varma MVS, Rodrigues AD. Quantitative Contribution of Six Major Transporters to the Hepatic Uptake of Drugs: "SLC-Phenotyping" Using Primary Human Hepatocytes. *J Pharmacol Exp Ther*. 2019;370(1):72-83. Epub 2019/04/13. doi: 10.1124/jpet.119.257600. PubMed PMID: 30975793.
5. Schiffer L, Barnard L, Baranowski ES, Gilligan LC, Taylor AE, Arlt W, Shackleton CHL, Storbeck KH. Human steroid biosynthesis, metabolism and excretion are differentially reflected by serum and urine steroid metabolomes: A comprehensive review. *J Steroid Biochem Mol Biol*. 2019;194:105439. Epub 20190727. doi: 10.1016/j.jsbmb.2019.105439. PubMed PMID: 31362062; PMCID: PMC6857441.
6. Maruti SS, Li L, Chang JL, Prunty J, Schwarz Y, Li SS, King IB, Potter JD, Lampe JW. Dietary and demographic correlates of serum beta-glucuronidase activity. *Nutr Cancer*. 2010;62(2):208-19. doi: 10.1080/01635580903305375. PubMed PMID: 20099195; PMCID: PMC2858007.
7. Chen Z, Peeters RP, Flach W, de Rooij LJ, Yildiz S, Teumer A, Nauck M, Sterenborg R, Rutten JHW, Medici M, Edward Visser W, Meima ME. Novel (sulfated) thyroid hormone transporters in the solute carrier 22 family. *Eur Thyroid J*. 2023;12(4). Epub 20230615. doi: 10.1530/ETJ-23-0023. PubMed PMID: 37074673; PMCID: PMC10305468.
8. Bush KT, Wu W, Lun C, Nigam SK. The drug transporter OAT3 (SLC22A8) and endogenous metabolite communication via the gut-liver-kidney axis. *J Biol Chem*. 2017;292(38):15789-803. Epub 20170801. doi: 10.1074/jbc.M117.796516. PubMed PMID: 28765282; PMCID: PMC5612110.
9. Wu W, Jamshidi N, Eraly SA, Liu HC, Bush KT, Palsson BO, Nigam SK. Multispecific drug transporter Slc22a8 (Oat3) regulates multiple metabolic and signaling pathways. *Drug Metab Dispos*. 2013;41(10):1825-34. Epub 20130806. doi: 10.1124/dmd.113.052647. PubMed PMID: 23920220; PMCID: PMC3781372.
10. Liu R, Hao J, Zhao X, Lai Y. Characterization of Elimination Pathways and the Feasibility of Endogenous Metabolites as Biomarkers of Organic Anion Transporter 1/3 Inhibition in Cynomolgus Monkeys. *Drug Metab Dispos*. 2023;51(7):844-50. Epub 20230414. doi: 10.1124/dmd.123.001277. PubMed PMID: 37059471.
11. Jarvinen E, Deng F, Kiander W, Sinokki A, Kidron H, Sjostedt N. The Role of Uptake and Efflux Transporters in the Disposition of Glucuronide and Sulfate Conjugates. *Front Pharmacol*. 2021;12:802539. Epub 20220113. doi: 10.3389/fphar.2021.802539. PubMed PMID: 35095509; PMCID: PMC8793843.
12. Schnabolk GW, Gupta B, Mulgaonkar A, Kulkarni M, Sweet DH. Organic anion transporter 6 (Slc22a20) specificity and Sertoli cell-specific expression provide new insight on potential endogenous roles. *J Pharmacol Exp Ther*. 2010;334(3):927-35. Epub 20100602. doi: 10.1124/jpet.110.168765. PubMed PMID: 20519554; PMCID: PMC2939676.
13. Brawand D, Soumillon M, Necsulea A, Julien P, Csardi G, Harrigan P, Weier M, Liechti A, Aximu-Petri A, Kircher M, Albert FW, Zeller U, Khaitovich P, Grutzner F, Bergmann S, Nielsen R, Paabo S, Kaessmann H. The evolution of gene expression levels in mammalian organs. *Nature*. 2011;478(7369):343-8. Epub 20111019. doi: 10.1038/nature10532. PubMed PMID: 22012392.

14. Neumann A, Kucukali F, Bos I, Vos SJB, Engelborghs S, De Pooter T, Joris G, De Rijk P, De Roeck E, Tsolaki M, Verhey F, Martinez-Lage P, Tainta M, Frisoni G, Blin O, Richardson J, Bordet R, Scheltens P, Popp J, Peyratout G, Johannsen P, Frolich L, Vandenberghe R, Freund-Levi Y, Streffer J, Lovestone S, Legido-Quigley C, Ten Kate M, Barkhof F, Strazisar M, Zetterberg H, Bertram L, Visser PJ, van Broeckhoven C, Sleegers K, group E-As. Rare variants in IFFO1, DTNB, NLRC3 and SLC22A10 associate with Alzheimer's disease CSF profile of neuronal injury and inflammation. *Mol Psychiatry*. 2022;27(4):1990-9. Epub 20220216. doi: 10.1038/s41380-022-01437-6. PubMed PMID: 35173266; PMCID: PMC9126805.
 15. Bianchi S, Stimpson CD, Bauernfeind AL, Schapiro SJ, Baze WB, McArthur MJ, Bronson E, Hopkins WD, Semendeferi K, Jacobs B, Hof PR, Sherwood CC. Dendritic morphology of pyramidal neurons in the chimpanzee neocortex: regional specializations and comparison to humans. *Cereb Cortex*. 2013;23(10):2429-36. Epub 20120808. doi: 10.1093/cercor/bhs239. PubMed PMID: 22875862; PMCID: PMC3767963.
 16. Kanton S, Boyle MJ, He Z, Santel M, Weigert A, Sanchis-Calleja F, Gujjarro P, Sidow L, Fleck JS, Han D, Qian Z, Heide M, Huttner WB, Khaitovich P, Paabo S, Treutlein B, Camp JG. Organoid single-cell genomic atlas uncovers human-specific features of brain development. *Nature*. 2019;574(7778):418-22. Epub 20191016. doi: 10.1038/s41586-019-1654-9. PubMed PMID: 31619793.
 17. Otani T, Marchetto MC, Gage FH, Simons BD, Livesey FJ. 2D and 3D Stem Cell Models of Primate Cortical Development Identify Species-Specific Differences in Progenitor Behavior Contributing to Brain Size. *Cell Stem Cell*. 2016;18(4):467-80. Epub 20160331. doi: 10.1016/j.stem.2016.03.003. PubMed PMID: 27049876; PMCID: PMC4826446.
-
1. Brawand D, Soumillon M, Necsulea A, Julien P, Csardi G, Harrigan P, Weier M, Liechti A, Aximu-Petri A, Kircher M, Albert FW, Zeller U, Khaitovich P, Grutzner F, Bergmann S, Nielsen R, Paabo S, Kaessmann H. The evolution of gene expression levels in mammalian organs. *Nature*. 2011;478(7369):343-8. Epub 20111019. doi: 10.1038/nature10532. PubMed PMID: 22012392.
 2. Jarvinen E, Deng F, Kiander W, Sinokki A, Kidron H, Sjostedt N. The Role of Uptake and Efflux Transporters in the Disposition of Glucuronide and Sulfate Conjugates. *Front Pharmacol*. 2021;12:802539. Epub 20220113. doi: 10.3389/fphar.2021.802539. PubMed PMID: 35095509; PMCID: PMC8793843.
 3. Wu W, Jamshidi N, Eraly SA, Liu HC, Bush KT, Palsson BO, Nigam SK. Multispecific drug transporter Slc22a8 (Oat3) regulates multiple metabolic and signaling pathways. *Drug Metab Dispos*. 2013;41(10):1825-34. Epub 20130806. doi: 10.1124/dmd.113.052647. PubMed PMID: 23920220; PMCID: PMC3781372.

Reviewers' Comments:

Reviewer #1:

Remarks to the Author:

The paper is improved and the Discussion clearer.

Two minor points that may be clarified with a few lines in the Discussion:

The authors now distinguish SLC22A8 from SLC22A10 because the former is a multi specific transporter that also transports sex steroids, which seems reasonable. This is also probably true of SLC22A20, which (although a pseudogene in humans) appears to be an odorant transporter as well as a sex steroid transporter. This should be clarified since the revised paper now leaves an impression that SLC22A20 stands apart as (only?) a sex steroid transporter.

In the older literature, a large amplified cluster of nonhuman (mouse) SLC22 transporters has been reported but not studied much. That cluster has remained confusing for the field since, by early sequence analysis, those murine transporters seemed similar to SLC22A10. Do the authors evolutionary analyses clarify this? If the authors are in a position to do so, it could be useful to the field.

Reviewer #2:

Remarks to the Author:

The authors have satisfactorily answered all the comments I made.

Reviewer #3:

Remarks to the Author:

The authors made considerable effort to provide additional data which convincingly strengthen their arguments. Thus I recommend publishing the manuscript as is

Reviewer #1 (Remarks to the Author):

The paper is improved and the Discussion clearer.

Two minor points that may be clarified with a few lines in the Discussion:

The authors now distinguish SLC22A8 from SLC22A10 because the former is a multi specific transporter that also transports sex steroids, which seems reasonable. This is also probably true of SLC22A20, which (although a pseudogene in humans) appears to be an odorant transporter as well as a sex steroid transporter. This should be clarified since the revised paper and now leaves an impression that SLC22A20 stands apart as (only?) a sex steroid transporter.

In the older literature, a large amplified cluster of nonhuman (mouse) SLC22 transporters has been reported but not studied much. That cluster has remained confusing for the field since, by early sequence analysis, those murine transporters seemed similar to SLC22A10. Do the authors evolutionary analyses clarify this? If the authors are in a position to do so, it could be useful to the field.

Response:

We agree with the reviewer that we should describe what was known about mouse *Slc22a20* in the discussion section. Though human SLC22A20 is a pseudogene, mouse *Slc22a20* (Oat6) encodes a functional transporter (see publications: JC Monte et al¹; GW Schnabolk et al.^{2,3}; and W Wu et al.⁴. In brief, these studies show a broad range of organic anion ligands of *Slc22a20* as well as its tissue localization in the mouse and phylogenetic analyses. All of these studies support the notion that *Slc22a20* shares similar substrates and inhibitors with SLC22A6 and SLC22A8, and may be more broadly selective than SLC22A24 and SLC22A9/10.

A recent paper describes the interesting, amplified cluster of mouse SLC22 transporters (OAT clade) as important for the uptake of thyroid hormone and sulfate conjugates, which is in contrast to the human organic anion transporters⁵. Interestingly, this study showed that several of the mouse SLC22 transporters, including, *Slc22a27*, *Slc22a28*, *Slc22a29*, *Slc22a30*, transport thyroid hormones (T3, T4) and exhibit an even greater fold uptake of the thyroid hormone sulfates (T3S, T4S). However, they showed no significant uptake of these metabolites in human SLC22A9, SLC22A24, and SLC22A10 (with P220), except T4S a substrate of SLC22A9. Phylogenetic tree analysis also showed that mouse *Slc22a20* is closer to mouse and human SLC22A6 and SLC22A8. In addition, the mouse OAT clade (*Slc22a19*, *Slc22a26*, *Slc22a27*, *Slc22a28*, *Slc22a29*, *Slc22a30*) is clustered closer to each other and further away from human SLC22A9, SLC22A24, and SLC22A10⁵.

Based on the information above, we have made the following modifications:

- Include *Slc22a20* in our phylogenetic tree analysis (**Figure 1A**) along with the mouse *Slc22* OAT clade (*Slc22a19*, *Slc22a26*, *Slc22a27*, *Slc22a28*, *Slc22a29*, *Slc22a30*). Our phylogenetic tree indicates that SLC22A10 is closer to SLC22A24, whereas the other known steroid conjugate transporters, SLC22A6, SLC22A8, and *Slc22a20*, and the mouse *Slc22* OAT clade (*Slc22a19*, *Slc22a26*, *Slc22a27*, *Slc22a28*, *Slc22a29*, *Slc22a30*) are clustered further away. We acknowledge that future studies are needed to test whether the SLC22A10-P220L variant transports thyroid hormone and conjugates.
- We have modified the Discussion to include the above references that indicate tissue expression patterns and substrate specificity of *Slc22a20* as well as a discussion of our phylogenetic analysis (Discussion section, paragraph 2). Revised **Figure 1A** is here.

Reference in this response to reviewer file:

1. Monte JC, Nagle MA, Eraly SA, Nigam SK. Identification of a novel murine organic anion transporter family member, OAT6, expressed in olfactory mucosa. *Biochem Biophys Res Commun.* 2004;323(2):429-36. doi: 10.1016/j.bbrc.2004.08.112. PubMed PMID: 15369770.
2. Schnabolk GW, Gupta B, Mulgaonkar A, Kulkarni M, Sweet DH. Organic anion transporter 6 (Slc22a20) specificity and Sertoli cell-specific expression provide new insight on potential endogenous roles. *J Pharmacol Exp Ther.* 2010;334(3):927-35. Epub 20100602. doi: 10.1124/jpet.110.168765. PubMed PMID: 20519554; PMCID: PMC2939676.
3. Schnabolk GW, Youngblood GL, Sweet DH. Transport of estrone sulfate by the novel organic anion transporter Oat6 (Slc22a20). *Am J Physiol Renal Physiol.* 2006;291(2):F314-21. Epub 20060214. doi: 10.1152/ajprenal.00497.2005. PubMed PMID: 16478971; PMCID: PMC2825707.
4. Wu W, Bush KT, Liu HC, Zhu C, Abagyan R, Nigam SK. Shared Ligands Between Organic Anion Transporters (OAT1 and OAT6) and Odorant Receptors. *Drug Metab Dispos.* 2015;43(12):1855-63. Epub 20150910. doi: 10.1124/dmd.115.065250. PubMed PMID: 26358290; PMCID: PMC4658493.
5. Chen Z, Peeters RP, Flach W, de Rooij LJ, Yildiz S, Teumer A, Nauck M, Sterenborg R, Rutten JHW, Medici M, Edward Visser W, Meima ME. Novel (sulfated) thyroid hormone transporters in the solute carrier 22 family. *Eur Thyroid J.* 2023;12(4). Epub 20230615. doi: 10.1530/ETJ-23-0023. PubMed PMID: 37074673; PMCID: PMC10305468.

Reviewers' Comments:

Reviewer #1:

Remarks to the Author:

The comments have been carefully addressed.